# Mechanochemical feedback control of dynamin independent endocytosis modulates membrane tension in adherent cells

Joseph Jose Thottacherry[1], Anita Joanna Kosmalska[2,3], Amit Kumar[4], Amit Singh Vishen [1,5], Alberto Elosegui-Artola[2], Susav Pradhan[4], Sumit Sharma[6], Parvinder P. Singh[6], Marta C. Guadamillas[7], Natasha Chaudhary[8,9], Ram Vishwakarma[6], Xavier Trepat [2,3,10], Miguel A. del Pozo [7], Robert G. Parton [8], Madan Rao[1,5], Pramod Pullarkat[4], Pere Roca-Cusachs [2,3] & Satyajit Mayor[1,11]

Plasma membrane tension regulates many key cellular processes. It is modulated by, and can modulate, membrane trafficking. However, the cellular pathway(s) involved in this interplay is poorly understood. Here we find that, among a number of endocytic processes operating simultaneously at the cell surface, a dynamin independent pathway, the CLIC/GEEC (CG) pathway, is rapidly and specifically upregulated upon a sudden reduction of tension. Moreover, inhibition (activation) of the CG pathway results in lower (higher) membrane tension. However, alteration in membrane tension does not directly modulate CG endocytosis. This requires vinculin, a mechano-transducer recruited to focal adhesion in adherent cells. Vinculin acts by controlling the levels of a key regulator of the CG pathway, GBF1, at the plasma membrane. Thus, the CG pathway directly regulates membrane tension and is in turn controlled via a mechano-chemical feedback inhibition, potentially leading to homeostatic regulation of membrane tension in adherent cells.

---

[1] National Centre for Biological Sciences (NCBS), Tata Institute of Fundamental Research (TIFR), Bellary Road, Bengaluru 560065, India. [2] Institute for Bioengineering of Catalonia (IBEC), Barcelona 08028, Spain. [3] University of Barcelona, Barcelona 08036, Spain. [4] Raman Research Institute, C. V. Raman Avenue, Bengaluru 560080, India. [5] Simons Centre for the Study of Living Machines, National Centre for Biological Sciences (NCBS), Bengaluru 560065, India. [6] CSIR - Indian Institute of Integrative Medicine, Jammu 180001, India. [7] Integrin Signalling Lab, Cell Biology & Physiology Program, Cell & Developmental Biology Area, Centro Nacional de Investigaciones Cardiovasculares Carlos III (CNIC), Madrid 28029, Spain. [8] University of Queensland, Institute for Molecular Bioscience and Centre for Microscopy and Microanalysis, St Lucia, QLD 4072, Australia. [9] Department of Biochemistry, Weill Cornell Medical College, New York, NY 10065, USA. [10] Centro de Investigación Biomédica en Red en Bioingeniería, Biomateriales y Nanomedicina (CIBER-BBN) and Institució Catalana de Recerca i Estudis Avançats (ICREA), Barcelona 08010, Spain. [11] Institute for Stem Cell Biology and Regenerative Medicine, Tata Institute of Fundamental Research (TIFR), Bengaluru 560065, India. Correspondence and requests for materials should be addressed to S.M. (email: mayor@ncbs.res.in)

L iving cells sense and use force for multiple functions like development[1], differentiation[2], gene expression[3], migration[4] and cancer progression[5]. Cells respond to changes in tension, passively by creating membrane invaginations/blebs[6–8], and actively by modulating cytoskeletal–membrane connections, mechanosensitive channels and membrane trafficking[4,9,10]. Membrane trafficking through endo–exocytic processes can respond to and modulate membrane tension[10]. While exocytosis acts to reduce plasma membrane tension as a consequence of increasing net membrane area, endocytosis could function to reduce membrane area and enhance membrane tension.

Membrane tension has long been shown to affect the endocytic process. Decreasing tension by the stimulation of secretion or addition of amphiphilic compounds increases endocytosis[11,12]. On the other hand, an increase in tension upon hypotonic shock[11] or as evinced during mitosis[12], results in a decrease in endocytosis. Although many studies suggest that endocytosis responds to changes in membrane tension, the specific endocytic mechanisms involved in these responses have not been elucidated.

We have recently shown that upon relaxing the externally induced strain on cells, tubule-like membrane invaginations termed 'reservoirs' are created[6]. This is purely a passive mechanical response of the plasma membrane following which cells deploy active cellular processes to resorb the excess membrane (cartoon: Fig. 1a).

Here we explore the nature of such active responses. We test the role of multiple endocytic pathways on modulation of membrane tension by three different approaches. In parallel, we utilize optical tweezers to measure membrane tension on modulating endocytosis. We find that subsequent to the passive membrane response, a clathrin-, caveolin- and dynamin-

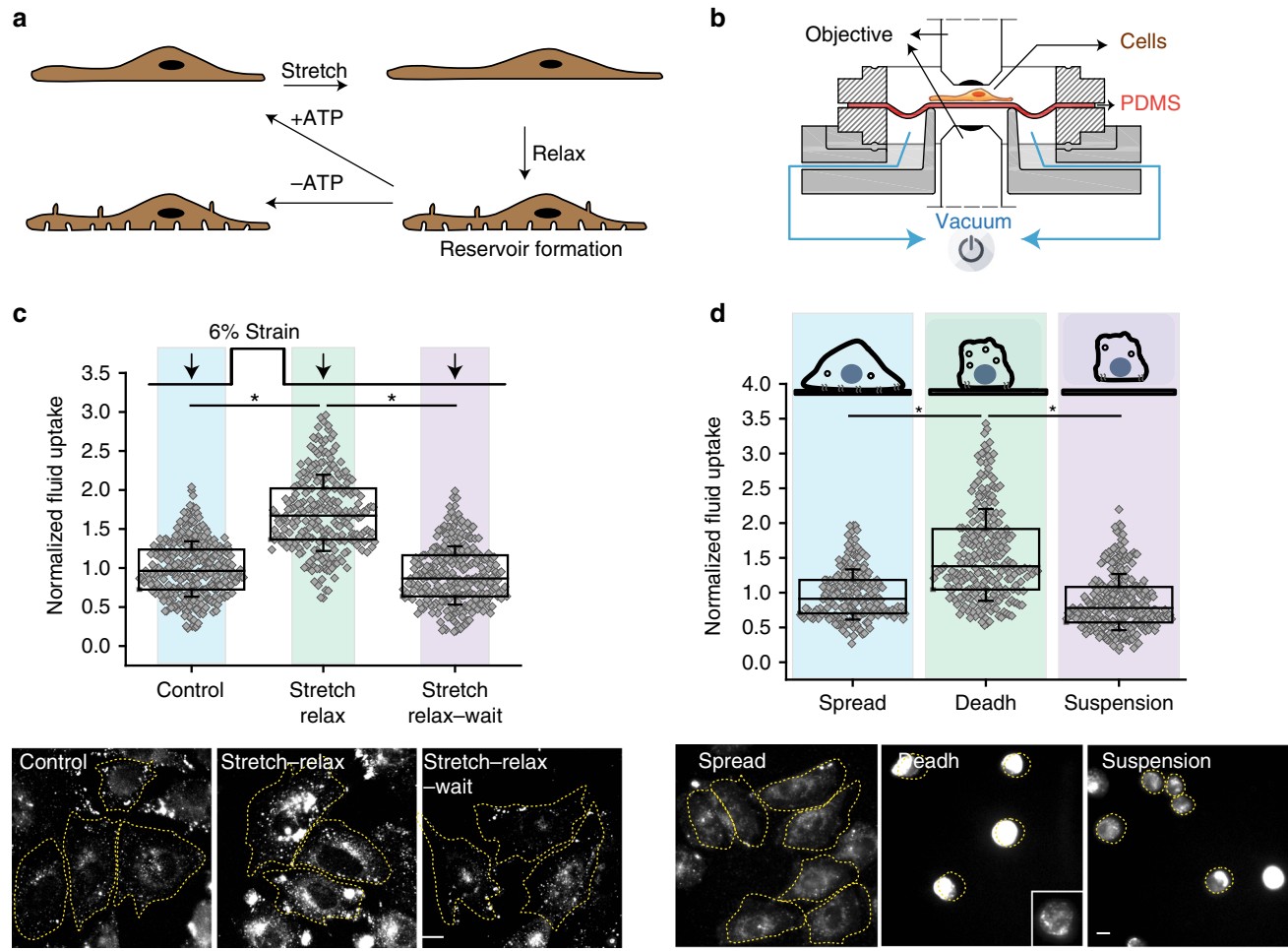

**Fig. 1** A fast transient endocytic response to decrease in membrane tension. **a** Cartoon showing membrane remodeling responses after mechanical strain. Cells after the stretch and relax protocol form invaginations termed 'reservoirs'[6]. These reservoirs are resorbed in a few minutes by an active process and requires ATP. **b** The illustration shows the longitudinal section of a vacuum-based equi-bi-axial stretching device. Cells plated on a PDMS sheet are stretched by the application of controlled vacuum below the circular PDMS sheet, which stretches it in a calibrated manner. Releasing the vacuum relaxes the strain on PDMS thus relaxing the cell. Cells plated on PDMS can be imaged in an upright or inverted microscope as required. **c** Fluid uptake (90 s) in CHO cells at steady state (steady state), immediately on relaxing the stretch (stretch–relax), or after a waiting time of 90 s on relaxing the stretch (stretch–relax–wait) (n = control (316), stretch–relax (257), stretch–relax–wait (277)). **d** Fluid uptake in CHO cells for 3 min in adhered cells (Spread), during de-adhering (Deadh), or immediately after cells are detached and in suspension (Suspension). Images and box plot show the extent of fluid-phase uptake under the indicated conditions (n = spread (196), deadh (241), suspension (274)). Box plot shows median, 25th and 75th percentile, and whiskers show the standard deviation. Individual data points are overlaid on box plot where each data point is the mean intensity per cell. The 'n' indicates total number of cells in each condition pooled from two different experiments with duplicates per experiment; *p < 0.001, ns not significant by Mann–Whitney U test. Scale bar, 10 μm

independent endocytic mechanism, the CLIC/GEEC (CG) pathway, is specifically and transiently upregulated. Vinculin, a protein involved in mechanotransduction[13], regulates this tension-mediated modulation of endocytosis in adherent cells. In its absence, the CG pathway fails to respond to changes in membrane tension and cell membrane tension is altered. On the other hand, perturbing the CG pathway directly modulates membrane tension, suggesting that this cellular mechanism is likely to be involved in homeostatic control of membrane tension.

## Results

**A rapid endocytic response to changes in membrane tension.** Active cellular processes are involved in resorbing the 'reservoirs' formed following a strain relaxation[6]. To determine whether endocytosis could be one such active process, we monitored the extent of endocytosis by providing a timed pulse of a fluid-phase marker, fluorescent-dextran (F-Dex), during and immediately after the stretch–relax procedure (using a custom built stretch device[6] shown in Fig. 1b). Compared to cells at steady state, there was a marked increase in fluid-phase endocytosis immediately after relaxation of the areal strain (Fig. 1c), while uptake was markedly reduced during strain application (Supplementary Fig. 1a). This increase in endocytosis was transient and disappeared as early as 90 s after strain relaxation (Fig. 1c). This also corresponds to the time scale of resorption of reservoirs by an active process observed earlier[6]. By rapidly upregulating endocytosis, cells thus respond to a net decrease in tension in a fast, transient fashion and return swiftly to a steady state.

Exocytosis delivers membrane rapidly in response to increased membrane tension during cell spreading[14]. On de-adhering, cells round off decreasing their surface area, while on replating, cells spread by adding membrane. Thus, we hypothesized that upregulation of endocytic pathways may help retrieve membrane on de-adhering due to a decrease in net membrane tension[8,15]. We monitored endocytosis of F-Dex during and immediately after the de-adhering and compared it to that measured in the spread state (Fig. 1d schematic). We found that the net fluid-phase uptake increased during de-adhering but subsided back to the steady-state level once de-adhered and held in suspension (Fig. 1d). The observed increase in endocytic uptake during the de-adhering process was not due to the accumulation of endocytosed cargo due to a recycling block (Supplementary Fig. 1b).

To further consolidate our findings, we used an alternate method to modulate membrane tension. We shifted cells from hypotonic to isotonic medium, which made passive invaginations similar to reservoirs called vacuole-like dilations (VLDs)[6]. This alternate method also results in an enhancement of fluid-phase endocytosis (Supplementary Fig. 1c). Together, these results suggested that using multiple strategies to reduce membrane tension triggered a fast and transient endocytic response.

**Response of multiple endocytic pathways to membrane tension.** A number of endocytic pathways function concurrently at the cell surface[16–19]. In addition to the well-characterized clathrin-mediated endocytic (CME) pathway, there are pathways that are independent of clathrin but utilize dynamin for vesicle pinching[18,20]. Additionally, there are clathrin- and dynamin-independent pathways which function in a number of cell lines[21–23], but not in all[24]. The CLIC/GEEC (clathrin-independent carrier/GPI-anchored protein-enriched early endosomal compartment) pathway is a clathrin- and dynamin-independent pathway, responsible for the internalization of a major fraction of the fluid phase and several glycophosphatidylinositol (GPI)-anchored proteins (GPI–AP)[21,23] as well as other plasma membrane proteins such as CD44[25]. To ascertain which of the multiple endocytic pathways respond to changes in tension, we examined specific cargo and regulators of these distinct pathways.

The endocytic uptake of the transferrin receptor (TfR), a marker of CME, did not increase in the cells which exhibited a transient rise in the fluid phase after a hypotonic shock (Fig. 2a) or detachment (Fig. 2b) as visualized using two -color fluorescence microscopy. However, uptake of the folate receptor, a GPI–AP that is internalized via the CG pathway[26,27], exhibited a considerable increase (Fig. 2c). This indicated that perhaps clathrin-independent endocytosis rather than CME might be involved in the fast response to a decrease in membrane tension.

To rule out clathrin-independent but dynamin-dependent endocytic pathways[18,21], we tested whether the increase in fluid-phase uptake required dynamin. We used a conditional triple knockout (TKO) cell line that removes all three dynamin isoforms from the genome[28], thereby abolishing all the dynamin-mediated endocytic pathways (Supplementary Fig. 2a). This dynamin TKO mouse embryonic fibroblasts (MEFs) exhibited higher steady-state fluid-phase endocytosis as reported earlier[28]. Despite this, TKO cells also transiently increased their fluid-phase endocytosis upon both stretch–relax cycles to the same extent as wild-type (WT) MEFs (Fig. 3a) and hypotonic/isotonic media changes (Supplementary Fig. 2b). Thus, neither CME nor dynamin-dependent endocytic pathways appear to respond to an acute reduction in membrane tension.

A caveolin-dependent endocytic process is important to retrieve specialized membrane on de-adhering[29], and a caveolae-mediated passive mechanism is reported to buffer the rapid increase in membrane tension and prevent cell lysis triggered by the flattening of caveolae[7,30]. To test if caveolin-dependent endocytic mechanisms could be important for this rapid endocytic upregulation, caveolin-null MEFs were subjected to the stretch–relax protocol. These cells exhibited a transient increase in fluid-phase uptake similar to their WT controls (Fig. 3a). In addition, caveolin-null cells also exhibit a fast transient upregulation of fluid-phase endocytosis during de-adhering as well (Supplementary Fig. 2c).

We next examined the morphology of the endocytic carriers formed by reduction of membrane tension induced by de-adhering using electron microscopy (EM). For this, we utilized cholera toxin bound horseradish peroxidase (CTxB-HRP), which is an excellent marker of the internalized plasma membrane. We used a procedure in which the surface remnant peroxidase reaction product is quenched with ascorbic acid, revealing only the internalized CTxB-HRP-labeled membrane[25]. After 5 min post de-adhering, the major endocytic structures labeled had the typical morphology of CG carriers (or CLICs) comprising structures with tubular and ring-shaped morphology (arrows, Supplementary Fig. 3a). Morphologically identical structures were also observed in WT MEFs at steady state[31] and in caveolin-null MEFs (arrows, Supplementary Fig. 3a) consistent with the observation of fast fluid-phase uptake in caveolin-null cells via CG (Supplementary Fig. 2c). At this time point, surface-connected caveolae (containing no peroxidase reaction product) persist in the WT MEFs (arrowheads, Supplementary Fig. 3a), consistent with the possibility that the caveolar pathway does not play a significant role in transiently modulating endocytosis at these early times of de-adhering.

To further understand the nature of endosomes on reducing tension, we imaged the fluid uptake in cells following a hypotonic shock at high resolution. Cells form larger endosomes on recovering from hypotonic shock (Supplementary Fig 3b) and the number of endosomes also increases (Supplementary Fig 3c, 3d and Supplementary Movie 1, 2).

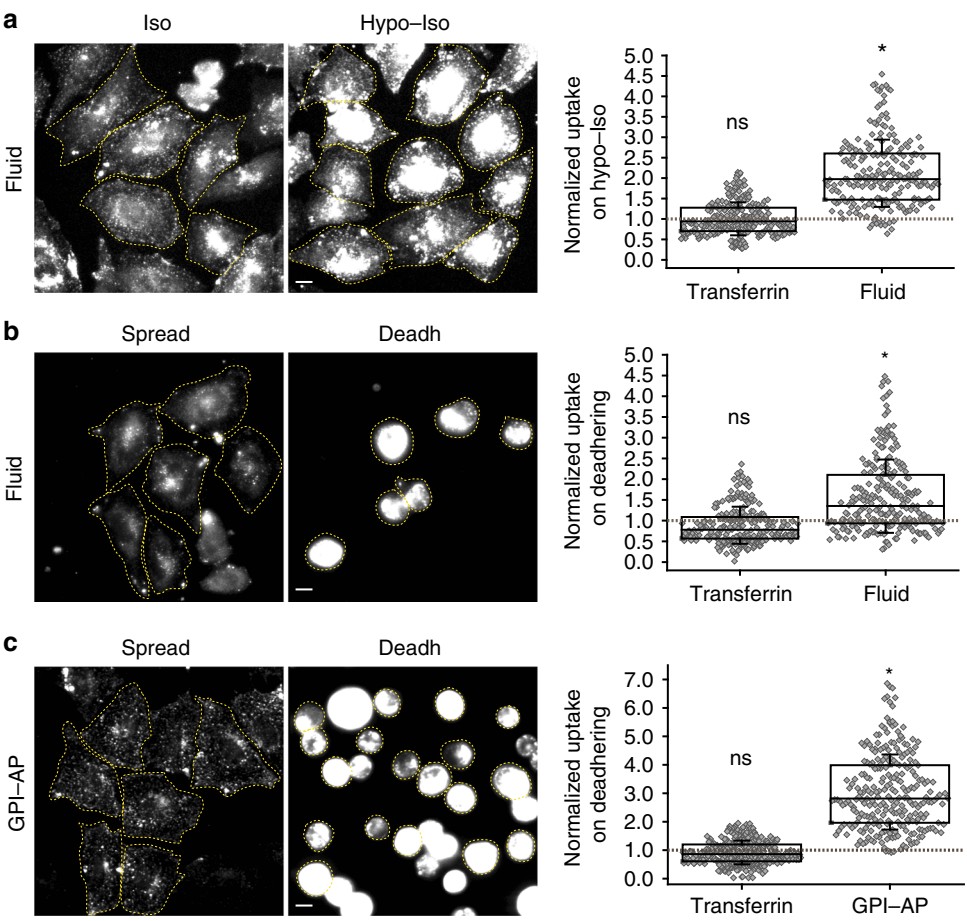

**Fig. 2** Endocytic pathways differ in their response to decrease in tension. **a** Fluid-phase and transferrin uptake in CHO cells under isotonic conditions (Iso) or immediately after shifting from the hypotonic to isotonic state (Hypo–Iso) by incubating cells with A647-Tf (Transferrin) or TMR-Dex (Fluid). Wide-field images (left) show the extent of endocytosed fluid-phase in isotonic or hypotonic-isotonic (Hypo–Iso) conditions. Box plot (right) show the extent of TMR-Dex and A647-Tf endocytosis in the Hypo–Iso condition normalized to those measured in the isotonic condition (gray dashed line) ($n =$ transferrin (266), fluid (214)). **b** Fluid-phase and transferrin uptake in CHO cells using TMR-Dex (Fluid) and A647-Tf (Transferrin) for 3 min when the cells are adherent (Spread) or during de-adhering (Deadh). Wide-field images (left) show the extent of endocytosed fluid-phase in Spread and during de-adhering condition (Deadh). Box plot (right) shows the extent of TMR-Dex and A647-Tf endocytosis in the de-adhered condition normalized to that measured in the Spread condition (gray dashed line) ($n =$ transferrin (246), fluid (244)). **c** GPI–AP and transferrin uptake in CHO cells using fluorescent folate to label GPI-anchored folate receptors (GPI–AP) and A647-Tf (Transferrin) in adherent cells (Spread) or during detachment (Deadh). Wide-field images (left) show the extent of endocytosed GPI-anchored folate receptor in Spread and during the de-adhering condition. Box plot (right) shows the extent of A647-Tf and folate receptor endocytosis in the de-adhered condition normalized to those measured in the Spread condition (gray dashed line) ($n =$ transferrin (321), GPI–AP (261)). Box plot shows median, 25th and 75th percentile, and whiskers show the standard deviation. Individual data points are overlaid on box plot where each data point is the mean intensity per cell. The '$n$' indicates total number of cells in each condition pooled from two different experiments with duplicates per experiment; *$p < 0.001$, ns not significant by Mann–Whitney $U$ test. Scale bar, 10 μm

Together, these experiments indicated that there is a rapid endocytic response that correlates with a reduction in membrane tension. This endocytic response is clathrin, dynamin and caveolin independent, and tracks the fluid-phase or GPI-anchored protein uptake which is endocytosed via the CG pathway. CG-mediated endocytosis is a high-capacity pathway capable of internalizing the equivalent of the entire plasma membrane area in 12 min[25], and of recycling a large fraction of endocytosed material[32]. Thus, the CG pathway could be involved in responding to tension by rapidly increasing endocytic capacity to endocytose a large portion of excess membrane for homeostasis.

**The CG pathway responds to alterations in membrane tension**. Since the CG cargo uptake negatively correlates with changes in tension, we explored this finding in further detail. The CG pathway is regulated by the small GTPases, ARF1, its guanine nucleotide exchange factor (GEF) GBF1, and CDC42 at the plasma membrane[27,33,34]. Hence, we utilized small-molecule inhibitors of CDC42 and GBF1 to acutely inhibit CG pathway[35,36]. The CDC42 inhibitor ML141 decreased fluid-phase endocytosis in cells at steady state but not CME (Supplementary Fig. 4a), and prevented the increase in fluid-phase uptake upon de-adhering (Supplementary Fig. 4b). Next, we utilized LG186, an inhibitor of GBF1, which also decreased fluid-phase endocytosis in cells at steady state but does not affect CME (Supplementary Fig. 4c). Inhibiting GBF1 prevents the increase in fluid-phase endocytosis observed upon stretch–relax (Fig. 3b) or de-adhering (Supplementary Fig. 4d). Similar to the decrease in fluid phase on increasing tension during stretch (Supplementary Fig. 1a), CD44, a CG pathway-specific cargo, shows reduced endocytosis during hypotonic shock (Supplementary Fig. 4e).

To confirm that this response is due to CG endocytosis, we assessed the effect of the stretch–relax protocol on cells that lack

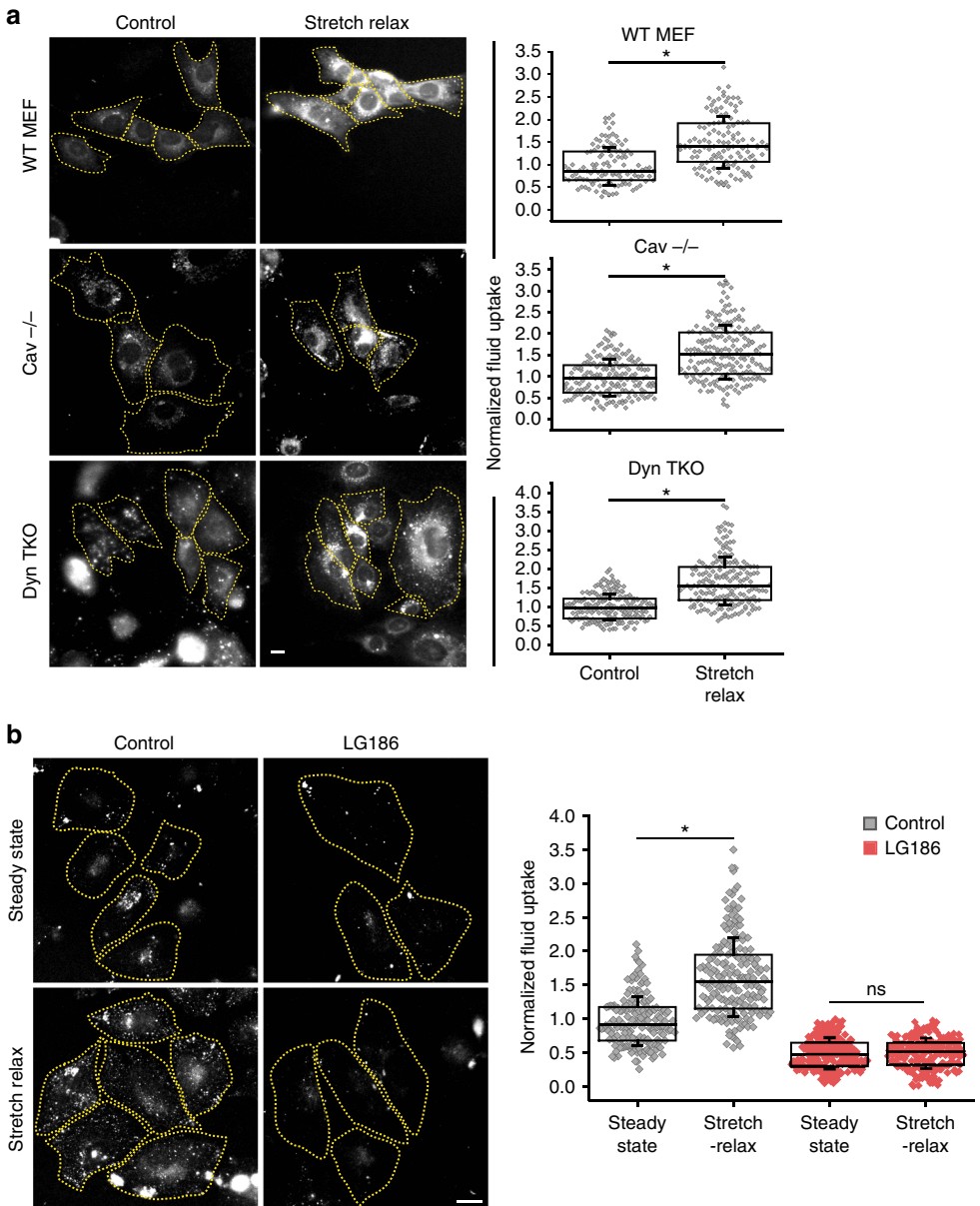

**Fig. 3** CG pathway is the primary pathway for fast endocytic response. **a** Fluid uptake in wild-type (WT MEF), caveolin-null (Cav$^{-/-}$) or conditional Dynamin triple knockout MEFs (Dyn TKO) for 90 s using TMR-Dex at steady state (control) and immediately after relaxing the stretch (stretch–relax). Images (left) show representative cells used to generate the box plots (right) which provide a quantitative measure of the extent of endocytosis of TMR-Dex for the indicated treatments. The uptake on stretch–relax is normalized to the steady-state uptake in the respective cell lines ($n$ = WT MEF–Control (117), Stretch–Relax (123); Cav$^{-/-}$–Control (173), Stretch–Relax (187); Dyn TKO–Control (179), Stretch–Relax (177)). **b** Fluid uptake in CHO cells treated with DMSO (Control) or with LG186 (10 μg/ml) (to inhibit GBF1) for 30 min, either at steady state (steady state) or immediately after relaxing the stretch (stretch–relax). Images (left) show representative cells used to generate the box plot (right) which provide a quantitative measure of the extent of endocytosis of TMR-Dex for the indicated treatments, normalized to the control steady-state condition ($n$ = Control–Steady state (170), Stretch–Relax (189); LG186–Steady state (248), Stretch–Relax (210)). Box plot shows median, 25th and 75th percentile, and whiskers show the standard deviation. Individual data points are overlaid on box plot where each data point is the mean intensity per cell. The '$n$' indicates total number of cells in each condition pooled from two different experiments with duplicates per experiment; *$p < 0.001$, ns not significant by Mann–Whitney $U$ test. Scale bar, 10 μm

CG endocytosis. HeLa cells lack a robust CG endocytic pathway[24,26] and use CME as the major endocytic pathway[24]. To confirm this, we downregulated CME with adaptor protein 2 (AP2) short hairpin RNA (shRNA)[37] in these cells. AP2 shRNA caused a decrease in both TfR endocytosis and fluid uptake in HeLa cells (Supplementary Fig. 5a). In contrast, AP2 shRNA treatment of AGS cells that exhibit a bonafide CG pathway[38] only reduced TfR uptake while fluid uptake was increased (Supplementary Fig. 5b). While the molecular basis for this defect is not

understood, we find that fluid-phase endocytosis in HeLa cells is insensitive to GBF1 inhibition by LG186 (Supplementary Fig. 5c). GBF1 is recruited to the plasma membrane as punctae in CHO cells[34] and in AGS[38] cells that exhibit robust CG endocytosis. However, Hela cells do not show an obvious recruitment of GBF1 to the plasma membrane compared to CHO cells (Supplementary Fig 5d/e). Correspondingly, the Hela cells did not exhibit a rapid increase in fluid-phase endocytosis on a hypotonic to isotonic shift (Supplementary Fig. 5f and 5g).

We recently found that IRSp53, an I-BAR domain protein, is necessary for the functioning of the CG pathway[38]. Cells lacking this protein show reduced fluid uptake compared to IRSp53-null cells restored with a wild-type version of IRSp53. Consistent with a central role for the CG pathway in the rapid tension-triggered endocytosis, lowering membrane tension in IRSp53-null cells failed to elicit the enhanced fluid-phase response, but the IRSp53-restored cells exhibited a robust enhancement (Supplementary Fig 6a).

A recent study showed that HeLa cells do respond to decreases in membrane tension in a GRAF1-dependent manner[39]. However, we note that this occurred under much more drastic conditions of membrane tension/osmolarity perturbation. In the aforementioned study, cells were treated with >6× diluted medium (<50 mOsm) for 10 min, followed by a 10 min return to isosmotic conditions. In contrast, in our method we subject cells to a maximum of 2× diluted medium (~150 mOsm) for 1 min, followed by a return to iso-osmolarity for a 1 min. Since the former treatment is an extreme hypotonic shock and for an extended period, we examined if similar to the HeLa, the CG-deficient IRSp53-null as well as the IRSp53-WT addback line would respond to this regime. We find that all the cell lines (with or without a bonafide CG pathway) respond to this extreme osmotic shock (Hypo 6×) followed by a return to isotonic conditions by enhancing their fluid-phase uptake (Supplementary Fig. 7a and b). These results suggest that extreme osmolarity changes are likely to trigger different mechanisms that may also function to restore membrane morphology, independent of the CG pathway. It should be emphasized that after the restoration period, the cells look distorted and in many cells the endosomes are present outside the bright-field image of the cell body (arrows, Supplementary Fig. 7a and 7b), indicating a distinct form of membrane internalization whose detailed mechanism needs further investigation.

In summary, the CG endocytic pathway is specifically involved in a rapid, transient response correlated to moderate changes in membrane tension.

**Passive and active response to changes in membrane tension**. As mentioned above, upon a rapid reduction in membrane tension, cells form passive structures such as reservoirs and VLDs similar to the response of an artificial vesicle. Reservoirs are formed upon strain relaxation in the membrane after stretching cells, whereas VLDs are formed by water expelled by the cell after a hypo-to-isotonic-shock recovery[6]. Both reservoirs and VLDs are reabsorbed and disappear within a couple of minutes, coincidental with an increase in endocytosis. This lead us to test if inhibiting the CG pathway could have a measurable impact on the rate of disappearance of such passive structures. The CG pathway exhibits exquisite temperature sensitivity and is barely functional at room temperature (25 °C), and is not efficient even at 30 °C unlike CME in CHO cells (Fig. 4a). Correspondingly, the reservoir resorption in CHO cells was impaired after lowering temperature (Fig. 4b). In addition, inhibition of the CG pathway in CHO cells by inhibiting GBF1 reduced the rate of reservoir reabsorption at 37 °C (Fig. 4a). In contrast, HeLa cells lacking a characteristic CG pathway did not show any difference in the rate of disappearance of reservoirs upon inhibiting GBF1 and were much less affected by lowering of temperature than CHO cells (Supplementary Fig. 8a). Thus, the resorption of the reservoirs is partly due to an increase in CG endocytosis.

We next examined if passively generated structures could help initiate endocytosis at the sites of their formation. The gradual disappearance of each reservoir indicates that it is not a single step process (Fig. 4b), and therefore reservoirs are unlikely to be pinched off directly as endosomes. Further, we did not observe endosomes form at the site of the reservoirs (Supplementary Fig. 9a). To test this, we took advantage of our earlier observation that cells plated on polyacrylamide gels do not form VLDs upon hypotonic to isotonic shifts[6]. Despite the lack of generation of VLDs in cells grown on polyacrylamide (Supplementary Fig. 9b), they still showed an increase in endocytosis similar to when plated on glass, upon exposure to hypo-to-isotonic-shock procedure (Supplementary Fig. 9c). Thus, VLD formation is not necessary for the endocytic response. Together, these data suggest that CG endocytosis occurs subsequent to the passive response by the membrane and the active response of the CG pathway does not depend on the passive morphological changes in the membrane exhibited by the cell.

**Role of the CG pathway in setting membrane tension**. Since the CG endocytic pathway responded to changes in membrane tension, we hypothesized that it might be involved in the setting of steady-state plasma membrane tension as well. To explore this hypothesis, we measured tether forces by pulling membrane tethers using optical tweezers in adherent cells[40]. The restoring force experienced by the bead associated with the membrane tethers provides a way to measure the plasma membrane tension[41] (Fig. 5a and Methods). Tether force is related to membrane tension by $T = F_0^2/8B\pi^2$ where $F_0$ is the tether force and $B$ is the bending stiffness (related to the force needed to bend a membrane of a given radius of curvature). Unlike in bare lipid vesicles, the membrane tension term from tether forces in cells is a combination of in-plane membrane tension and membrane–cytoskeleton adhesion. It is difficult to separate out these contributions and therefore the restoring force measured by these optical trap experiments is referred to as apparent membrane tension or effective membrane tension[4,42].

We found that acutely inhibiting the CG pathway by GBF1 inhibition drastically reduced tether forces in adherent cells at steady state (Fig. 5b). We next examined tether forces in cells where the CG pathway is upregulated. We reasoned that since the Dynamin TKO cells show a higher fluid-phase endocytosis (Fig. 5c and Supplementary Fig 9d), it is likely that this would increase its effective membrane tension. Tether forces were indeed higher in the Dynamin TKO cells compared to control cells (Fig. 5d). Consistent with the role of the CG pathway in setting effective membrane tension, inhibiting the CG pathway in Dynamin TKO cells by GBF1 inhibition (Fig. 5c and Supplementary Fig 9d) reduced the tether forces below control levels (Fig. 5d).

To further confirm this observation, we measured tether forces on acutely increasing CG endocytosis by using Brefeldin A (BFA) as reported earlier[33]. BFA treatment disrupts endoplasmic reticulum (ER) to Golgi secretion and also serves to free up ARF1, making it available at the cell surface to increase CG endocytosis[33]. This increase is mediated through GBF1-sensitive CG endocytosis (Fig. 5e). We treated cells with BFA and measured tether forces when the increase in endocytosis was most prominent. Tether forces were higher on treating cells with BFA compared to the control case (Fig. 5g). Since BFA treatment also inhibits secretion[43], this could also increase the effective membrane tension due to a reduction of membrane delivery from the secretory pathway, independent of its effect on CG endocytosis. To test this, we treated HeLa cells that lack the CG pathway with BFA. BFA treatment disrupted the Golgi in both CHO and HeLa cells (Supplementary Fig. 9e), consistent with its inhibition of the secretory pathway. However, neither fluid-phase uptake (Fig. 5f) nor the tether forces were affected in HeLa cells (Fig. 5g). This indicated that the increase in tether force in CHO

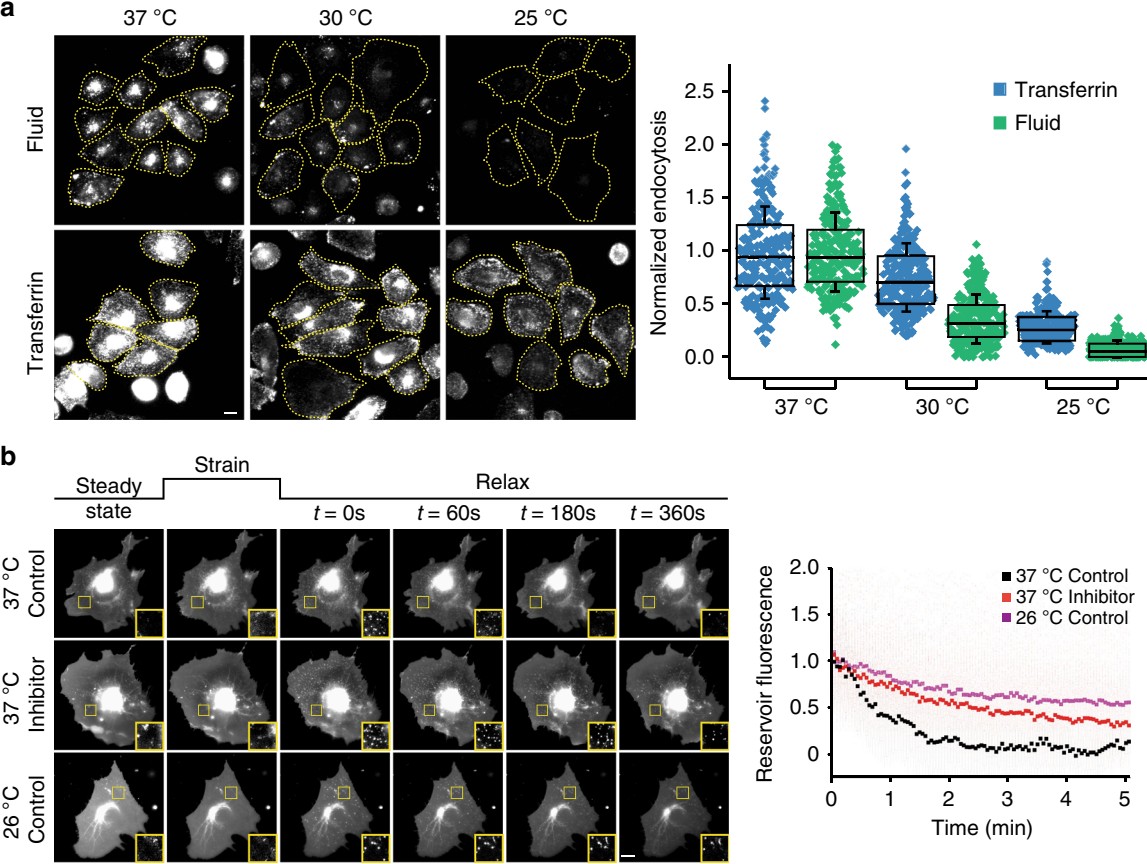

**Fig. 4** Temperature dependence of CG pathway and reservoir resorption. **a** Fluid and transferrin uptake in CHO cells (pre-equilibrated at the indicated temperatures) using TMR-Dex (Fluid) and Tf-A647 (Transferrin) for 5 min at the respective temperatures. All values were normalized to the respective mean endocytosis at 37 °C. Representative images (left) of cells used to generate the box plot (right) were obtained from two different experiments with duplicates per experiment (total number of cells = Transferrin: 37 °C (252), 30 °C (269), 25 °C (252); Fluid: 37 °C (249), 30 °C (291), 25 °C (270)). Box plot shows median, 25th and 75th percentile, and whiskers show the standard deviation. Individual data points are overlaid on box plot where each data point is the mean intensity per cell. The 'n' indicates total number of cells in each condition pooled from two different experiments with duplicates per experiment. Scale bar, 10 μm. **b** The reservoir fluorescence intensity after stretch–relax of CHO cells transfected with a fluorescent membrane marker (pEYFP-mem) was quantified as a function of time at 37 °C in the absence (37 °C control) or presence of LG186 (37 °C inhibitor), or at room temperature (26 °C control). Each point represents mean reservoir intensity over time from more than 100 reservoirs from at least 10 cells. Scale bar, 10 μm

cells upon BFA treatment is specifically due to an increase in CG endocytosis at these timescales.

These results show that modulating the CG pathway by activating or inhibiting key regulators modifies the membrane tension.

**Mechanical manipulation of the CG endocytosis machinery**. We next tested if key regulatory molecules involved in different endocytic pathways could be directly modulated by changes in tension. GBF1 is involved in the CG pathway and re-localizes from the cytosol to distinct punctae at the plasma membrane upon activation as visualized using total internal reflection fluorescence (TIRF) microscopy[33,34]. We imaged GBF1–GFP recruitment to the plasma membrane in live cells, during a hypotonic shock and after recovery, using TIRF microscopy. GBF1 punctae were lost on hypotonic shock (Fig. 6a, b) indicating a direct response by GBF1 on increasing tension. On the other hand, recovery from a hypotonic shock caused the rapid assembly of GBF1 punctae (Fig. 6a, b). In contrast, clathrin, which redistributes from the cytosol to membrane to help in CME, is not affected by similar changes in tension (Supplementary Fig. 10a). These experiments indicated that the molecular

machinery involved in regulating the CG pathway is modulated by moderate changes in membrane tension.

**Vinculin serves as a mechanotransducer**. For cells to respond to changes in tension, they must first sense and transduce this information. Since focal adhesion-related molecules help transduce and respond to force[5,44,45] we hypothesized that these molecules could transduce a mechanical stimulus to regulate endocytic processes. Membrane tension could be transmitted across integrins[46] and even regulate focal adhesion positioning through vinculin[47]. Indeed, several of these proteins were 'hits' in a recent RNA interference screen for genes that influence CG endocytosis[48]. The focal adhesion is an intricate macromolecular complex that has multiple functional modules[49] of which vinculin is a critical part of the mechanotransduction machinery[44,45,49]. Unlike the 'hits', Talin or p130CAS, there is only a single functional isoform of vinculin in non-muscle cells. Therefore, we used vinculin-null MEFs[50] to test its role in the mechano-responsive behavior of the CG pathway.

We noticed that the basal fluid-phase uptake in vinculin-null MEFs is higher than that of WT MEFs (Supplementary Fig. 10b). To directly test if vinculin could be involved in the tension-

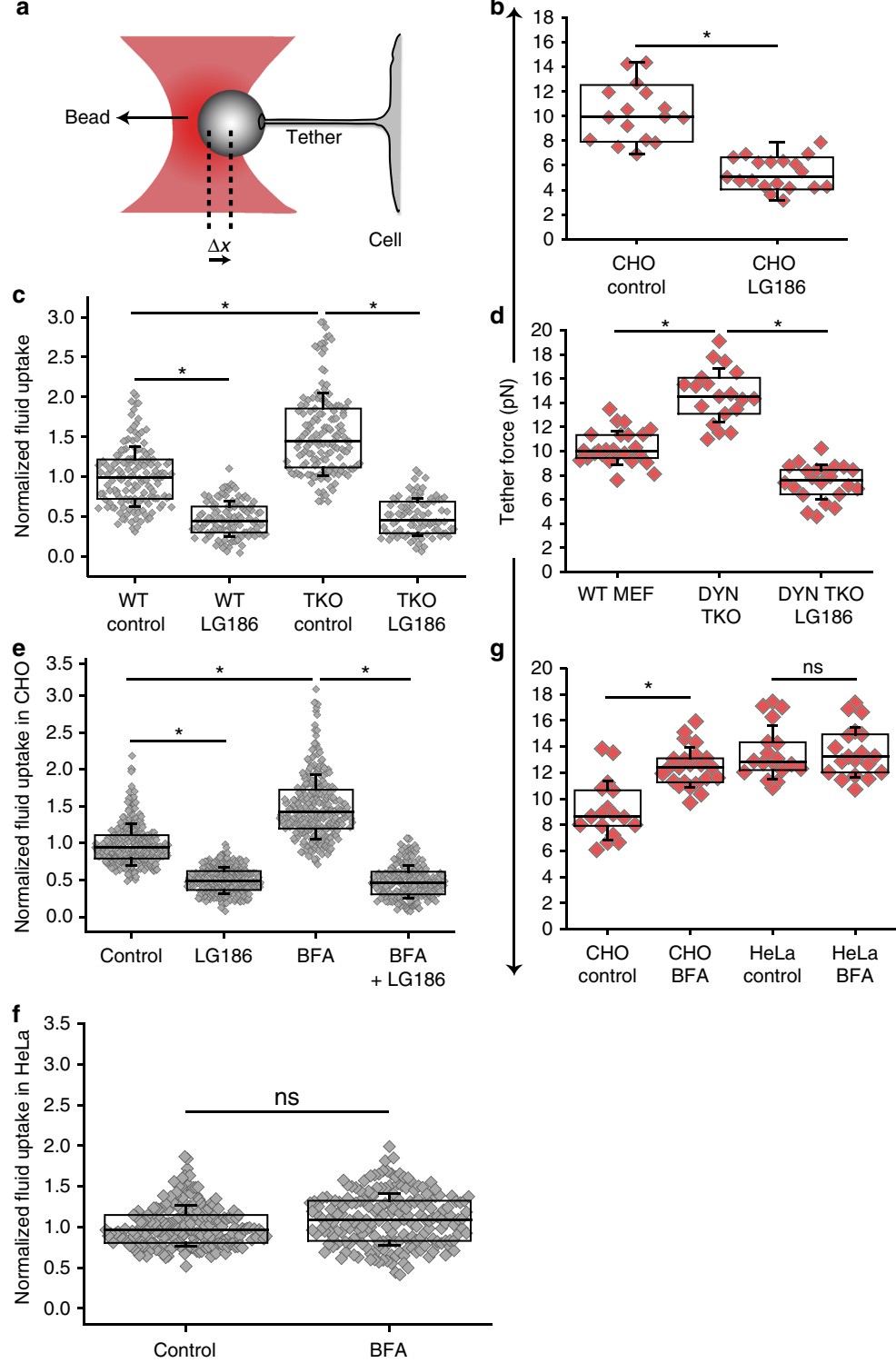

sensitive regulation of the CG pathway, we stretched vinculin-null cells. Unlike WT MEFs that show ~82% reduction in uptake on stretching (Supplementary Fig. 1a), vinculin-null MEFs show only a moderate drop of ~36% at the same strain levels (Fig. 7a). Increasing the extent of membrane tension by hypotonic shock showed a concomitant decrease in fluid-phase endocytosis of WT cells (Fig. 7b). By contrast, vinculin-null MEFs were much more refractory to the same extent of hypotonic shock, registering

a reduction in fluid-phase uptake only at much higher hypotonicity (Fig. 7b).

Furthermore, upon strain relaxation, fluid-phase endocytosis in vinculin-null MEFs did not show an increase (Fig. 7a), unlike that observed for WT MEFs (Fig. 3a) or CHO cells (Fig. 1c). However, the membrane of vinculin-null MEFs did respond to changes in membrane tension (Supplementary Fig. 10c). This was further confirmed in the de-adhering assay where vinculin-

**Fig. 5** CG pathway regulates membrane tension. **a** Cartoon shows a membrane tether attached to a polystyrene bead trapped in an optical trap, used to measure tether forces. Displacement of the bead from the center of the trap ($\Delta x$) gives an estimate of the tether force (F) of the cell (see Methods). **b** Tether forces from CHO cells either treated with DMSO (CHO Control) or LG186 (CHO LG186) for 30 min. The box plot shows data points, with each point corresponding to a tether per cell with data combined ($n = 16$ (CHO control) and 19 (CHO LG186)) from two different experiments. **c** Fluid uptake in wild-type (WT) MEF or conditional Dynamin TKO cells either pre-treated with DMSO control or LG186. The box plot show fluid-phase uptake normalized to that observed in untreated WT MEF cells ($n =$ WT–Control (148), LG186 (110); TKO–Control (155), LG186 (87)). **d** Tether forces in WT MEF or conditional Dynamin TKO cells either pre-treated with DMSO or LG186 ($n = 25$ (WT MEF), 19 (DYN TKO) and 22 DYN TKO LG186)). **e** Fluid uptake in CHO cells treated with DMSO (Control) or with BFA (20 μg/ml) alone or with LG186 for 30 min ($n =$ Control (309), LG186 (319), BFA (290), BFA+LG186 (247)). **f** Fluid uptake in HeLa cells treated with BFA or DMSO control. The box plot shows the extent of fluid-phase uptake under the indicated conditions, normalized to that observed in control ($n =$ Control (207), BFA (193)). **g** Box plot shows tether forces measured in CHO or HeLa cells treated with DMSO (Control) or with BFA for 45 min ($n = 17$ (CHO Control), 23 (CHO BFA),18 (HeLa Control), 19 (HeLa BFA)). Box plot shows median, 25th and 75th percentile, and whiskers show the standard deviation. Individual data points are overlaid on box plot where each data point is the mean intensity per cell (**c**, **e**, **f**) or tether force per cell (**b**, **d**, **g**). The '$n$' indicates total number of cells in each condition pooled from two different experiments with duplicates per experiment; *$p < 0.001$, ns not significant by Mann–Whitney $U$ test

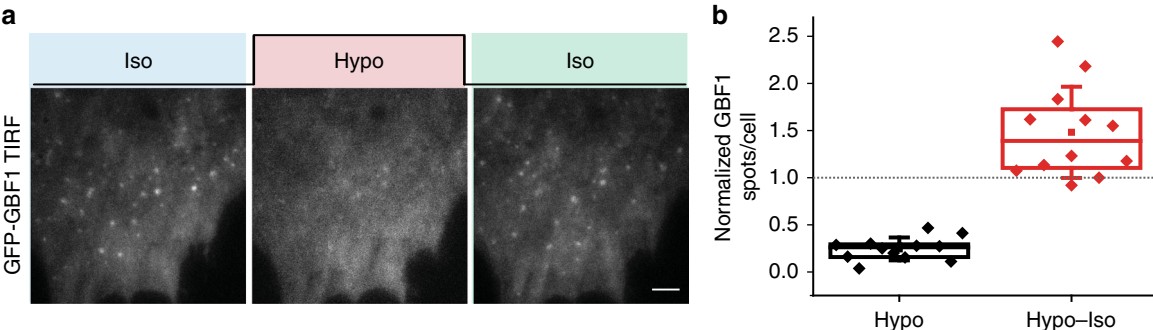

**Fig. 6** Mechanical modulation of CG molecular machinery. **a** GBF1–GFP punctae in WT MEF cells by live TIRF microscopy on modulating osmolarity by changing the media from isotonic (Iso) to 40% hypotonic (Hypo) and back to isotonic (Iso). **b** Quantification of the number of punctae per cell in (**a**). The GBF1 spots upon hypotonic shock and subsequent shift to isotonic medium are normalized to original number of spots in the respective cell and plotted as a box plot. Each data point is a measurement from a single cell and box plot shows data of 12 cells from two independent experiments. Scale bar, 10 μm

null cells did not show an increased fluid-phase uptake, unlike the WT cells (Supplementary Fig. 10d). Thus, vinculin-null cells do not exhibit an endocytic response to changes in tension. Further, when WT cells are de-adhered from the substrate and kept in suspension, they too failed to exhibit an endocytic response to changes in tension created during hypo–iso conditions (Supplementary Fig. 10e). However, similar to spread cells (Fig. 4b), the suspension cells also exhibit characteristics consistent with the passive changes expected from the reduction in membrane tension (Supplementary Fig. 10f and Supplementary Movie 3). These observations indicate that attachment to the surface via integrins, and concomitant vinculin activation, is important for the control of the endocytic response.

**Vinculin negatively regulates CG endocytosis.** To test if the endocytic effects of vinculin-null cells are specifically due to a lack of vinculin, we expressed full-length vinculin (Vin WT)[51] in vinculin-null cells. This caused a decrease in fluid-phase endocytosis (Fig. 7c and Supplementary Fig. 11a) and rescued the transient endocytic response on decrease in tension (Fig. 7c).

Vinculin is activated by binding to talin localized at integrin-mediated focal complex[51]. Unlike WT-Vinculin addback in the vinculin-null cells (Fig. 7c), vinculin with talin binding mutation (Vin-A50I)[51] did not reduce fluid uptake (Fig. 7d), indicating that talin is required for vinculin to regulate the CG pathway. On the other hand, constitutively active vinculin (Vin-CA) and Vin-CA with a talin binding mutation (Vin-A50I-CA)[51] reduced fluid-phase uptake significantly, but failed to respond to changes in tension (Fig. 7e). This indicates that talin-mediated activation of

vinculin is required for the mechano-response of the CG pathway.

Vinculin-null cells have a higher basal endocytosis rate (Supplementary Fig. 10b) and it is possible that they are unable to respond to a decrease in membrane tension and increase in endocytosis. Firstly, we confirmed that a GBF1-sensitive CG pathway is functional in vinculin-null cells. Upon GBF1 inhibition in vinculin-null cells, fluid-phase uptake decreased to the same levels as cells expressing vinculin, also suggesting that GBF1 operates downstream of vinculin (Supplementary Fig. 11a). Next, we observed that BFA-treated vinculin-null cells showed an increase in their endocytic rate which is sensitive to GBF1 inhibition, similar to WT cells (Supplementary Fig. 11b). This shows that vinculin-null cells have the potential to upregulate the CG pathway and their inability to do so on reducing tension is due to the lack of mechanotransduction.

We next tested if GBF1 shows a tension-dependent membrane localization in vinculin-null cells. The level of punctae remained constant and failed to respond to hypotonic shock (Fig. 7f), unlike that observed in WT MEF (Fig. 6a). The density of GBF1 punctae at the plasma membrane was also slightly higher in vinculin-null cells compared to WT cells (Supplementary Fig. 11c). This is consistent with higher fluid-phase endocytosis in vinculin-null cells compared to control MEF cell line (Supplementary Fig. 10b).

Finally, we determined the steady-state membrane tension in vinculin-null cells compared to WT cells. Tether forces, as measured using optical tweezers, were higher for vinculin-null cells compared to WT cells (Fig. 8a). The high tether force in cells lacking vinculin was drastically reduced on inhibiting the CG pathway (Fig. 8a), consistent with the role of the CG pathway in

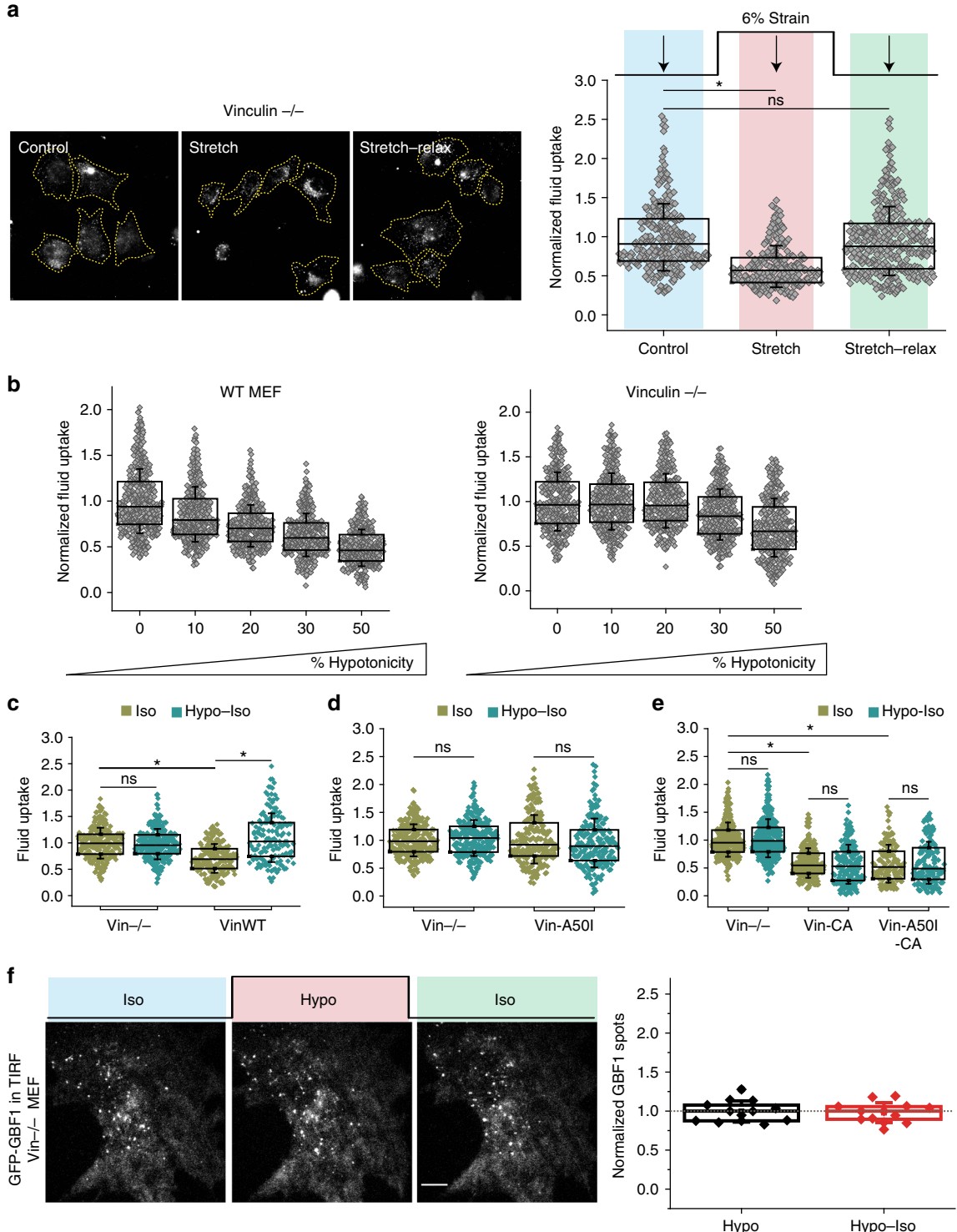

regulating the effective membrane tension. Thus, vinculin acts as a negative regulator of CG pathway and is necessary for the transduction of physical stimuli for the biochemical control of the CG pathway.

**Mechanochemical control of cell membrane tension**. To rationalize our experimental results, we propose a simple feedback inhibition model based on a mechanochemical mechanism for the sensing and control of membrane tension. The

mechanochemical mechanism involves (i) membrane flux via CG endocytosis, (ii) concentration of membrane-bound vinculin (in the active open configuration) and (iii) concentration of activated membrane-bound GBF1, which together go to sense and regulate the membrane tension. Membrane tension is determined by the relative balance between CG endocytosis and exocytosis, which in turn depends on the levels of activated GBF1 and tension. The levels of activated GBF1 depend on the concentration of membrane-bound vinculin. Such a negative feedback inhibition model can give rise to a robust control of membrane tension

**Fig. 7** Vinculin-dependent mechanoregulation of CG pathway. **a** Fluid uptake in vinculin-null cells either during a 6% stretch or on relaxing this strain (n = Control (281), Stretch (229), Stretch–Relax (347)). **b** Fluid uptake in WT and vinculin-null MEFs in increasing hypotonic medium as indicated (n = WT MEF: 0 (425), 10(391), 20 (416), 30 (368), 50 (346); Vinculin −/−: 0 (355), 10 (376), 20 (333), 30 (342), 50 (340)). **c** Fluid uptake in vinculin-null cells (Vin−/−) or Vin −/− transfected with Vinculin WT (VinWT) either in isotonic medium (Iso) or in isotonic medium after a hypotonic shock for 1 min (Hypo–Iso) (n = Vin−/−–Iso (256), Hypo-Iso (272); VinWT–Iso (188), Hypo-Iso (146)). **d** Fluid uptake in Vin −/− or Vin −/− transfected with vinculin with talin binding mutation (Vin-A50I) either in Iso or Hypo–Iso (n = Vin−/−–Iso (251), Hypo-Iso (264); Vin-A50I-Iso (219), Hypo_Iso (190)). **e** Fluid uptake in Vin −/−, Vin −/− transfected with constitutively active vinculin (Vin-CA) or constitutively active vinculin with talin binding mutation (Vin-A50I-CA) in Iso or Hypo-Iso (n = Vin −/−–Iso (327), Hypo-Iso (334); Vin-CA-Iso (238), Hypo-Iso (179); Vin-A50I-CA–Iso (139), Hypo-Iso (150)). Box plot shows median, 25th and 75th percentile, and whiskers show the standard deviation. Individual data points are overlaid on box plot where each data point is the mean intensity per cell. The 'n' indicates total number of cells in each condition pooled from two different experiments with duplicates per experiment; *p < 0.001, ns not significant by Mann–Whitney U test. **f** Vinculin-null cells transfected with GBF1–GFP and imaged live using TIRF microscopy on changing media from isotonic (Iso) to 40% hypotonic (Hypo) and back to isotonic (Iso). GBF1 organization at the plasma membrane during the osmotic shifts is shown in a representative cell (left panel). Number of punctae per cell during hypotonic and isotonic shifts is normalized to the initial number of spots (gray dotted line) and plotted as a box plot. Each data point is measurement from a single cell and box plot shows data of 13 cells from two independent experiments. Scale bar, 10 μm

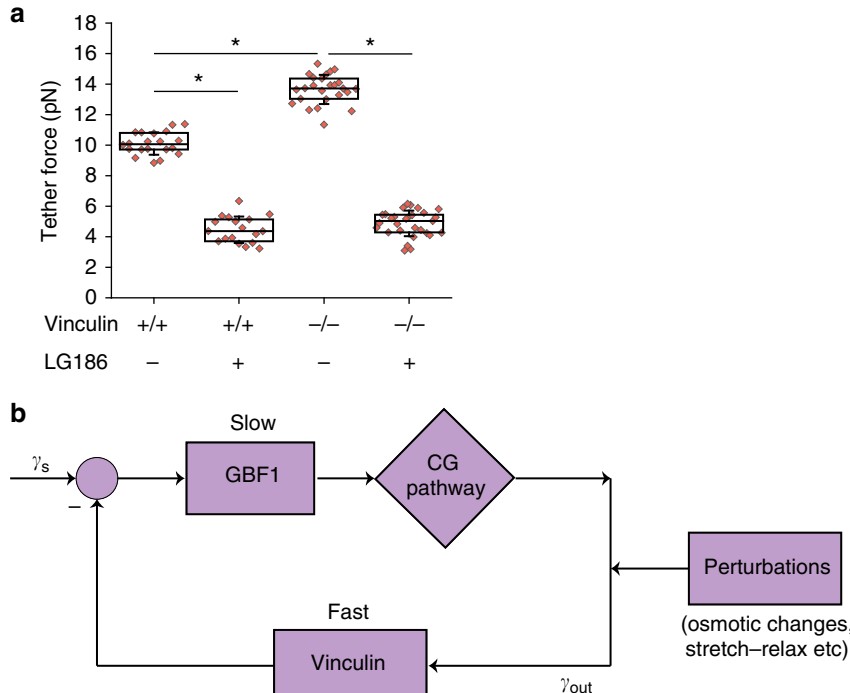

**Fig. 8** Membrane tension and vinculin. **a** Tether forces in WT (Vin +/+) or vinculin-null MEFs (Vin −/−) treated with LG186 (to inhibit GBF1-mediated CG pathway) compared to the control treated cells (total number of cells = 20 (Vin +/+), 25 (Vin +/+ with LG186), 25 (Vin −/−) and 29 (Vin −/− with LG186)). Vinculin-null cells show a higher basal membrane tension compared to WT MEF, while inhibiting the CG pathway drastically reduced membrane tension in both cell lines. Box plot shows median, 25th and 75th percentile, and whiskers show the standard deviation. Individual data points are overlaid on box plot where each data point is the tether force per cell. The 'n' indicates total number of cells in each condition pooled from two different experiments; *p < 0.001 by Mann–Whitney U test. **b** Feedback inhibition description provides a model for robust feedback control of membrane tension that involves slow activation and fast inhibition. To provide robust feedback control, tension set point $\gamma_s$ should be compared to the instantaneous tension $\gamma_{out}$ and compensate for the difference. See supplementary information (Supplementary Note 1) for detailed description of the mechanochemical model

(Fig. 8b). The mathematical details of this model are discussed in the supplementary information (Supplementary Note 1).

## Discussion

Membrane tension has been long proposed to be tightly coupled to vesicular trafficking through endo–exocytic pathways. However, the specific trafficking mechanisms have remained elusive. Here, we show that a GBF1-, ARF1-, CDC42- and IRSp53-dependent CG endocytosis rapidly responds to changes in tension, and helps restore any change from a tension set point (Fig. 8b). When membrane tension decreases (increases) in cells attached to a surface, there is a correlated increase (decrease) in the CG pathway, bringing about a rapid endocytic response to reset the cell's resting membrane tension. Conversely, inhibiting the CG pathway decreases membrane tension while upregulating the pathway increases membrane tension. This negative feedback inhibition is used in many different biological contexts to maintain homeostasis[52], and is captured by our simple feedback inhibition model. The separation of time scales between the fast dynamics of the activation of vinculin (triggered by sensing talin stretch), and slower dynamics of the displacement of GBF1 from the membrane and inhibition of CG endocytosis, suggests that vinculin is a sensor

of the instantaneous membrane tension. The negative feedback inhibition is predicated on a fast sensing of tension by vinculin and a slower regulation of the endocytic machinery.

Akin to the passive response to tension[6], physical parameters could directly regulate the active endocytic machinery by influencing the extent of membrane deformation needed to make an endocytic vesicle. However, our results from studying vinculin-null cells suggest otherwise. Vinculin, a key focal adhesion protein, transduces many mechanical inputs at the site of the focal adhesion into information for the cell to process[13,45,49]. In this context, it appears that vinculin plays a central role in transducing the increase (or decrease) in membrane tension to the CG pathway to help inhibit (or activate) its endocytic mechanism (Supplementary Fig. 12a). Membrane tension has been shown to regulate tension that is experienced by integrin molecule[53], and tension also influences focal adhesion positioning[47]. In turn, this focal adhesion tension, transduced via vinculin, regulates GBF1 and thus the CG pathway, further influencing membrane tension. Thus, it is likely that the changes in membrane tension are communicated via changes experienced in forces at attachment sites. Consistent with this hypothesis, cells in suspension that lack attachment sites do not modulate their CG pathway, similar to cells that lack vinculin.

Vinculin recruits phosphatidylinositol-3-kinase (PI3K) in a force-dependent manner helping in tumor progression[54] and mechanical cues transmitted through focal adhesions have been implicated in cancer progression[5]. PI3K products help recruit GBF1, an ARF1 GEF, to the plasma membrane and this is necessary for CG endocytosis[55], indicating a possible mechanism for vinculin-dependent regulation of the CG pathway. Regardless of the details, mechanotransduction mediated by vinculin is important for translating mechanical information into a biochemical read-out to influence the CG endocytic rate.

Multiple endocytic pathways operate in a cell with a possible functional specialization for each. Caveolae passively buffer increases in tension[7], while CME concentrates specific ligands and mediates robust endocytosis despite the increase in tension[56]. Membrane tension is also involved in transition of clathrin coats from flat to curved[57], while actin machinery helps provide forces to internalize clathrin coat under high tension[56,58,59]. De-adhered cells exhibit an increased caveolin-mediated internalization that persists over hours in suspension, and is crucial for anchorage-dependent growth and anoikis[29]. Unlike the caveolar pathway, we find that the CG pathway showed an upregulation of endocytosis only during de-adhering that did not persist in suspension. The CG pathway is also more temperature-sensitive than CME. In cells such as HeLa that lack a typical dynamin-independent CG pathway[24,26], they appear to have an AP2-, GRAF1- and dynamin-dependent endocytic machinery[24]. These cells also do not upregulate endocytosis in response to moderate osmotic shock but do so in response to extreme osmolarity changes applied for longer time points (such as Hypo6X-Iso). Considering that osmotic changes are also likely to trigger different cellular responses[60], the endocytic response in these contexts while dependent on GRAF1[39] requires further characterization. Inhibition of GRAF1 in HeLa causes a blebbing response[39] (possibly due to a complete shutdown of endocytosis[8]) and this helps in cancer cell migration[39]. Together, these results indicate that different endocytic pathways have distinct regimes of operation and serve important functions in eukaryotic cells.

In this study we find that the composition-sensitive high-capacity CG pathway[25,32] is modulated by mechanochemical inputs. This could increase the potential of membrane tension to regulate other cellular processes, such as those that may need a supply of membrane at long and short time scales, e.g. migration

and phagocytosis. Thus, the CG pathway responds to and coordinates a variety of cellular inputs, including membrane tension, and is likely to function in multiple physiological contexts.

## Methods

**Cell lines, constructs and synthesis of LG186**. See Supplementary Materials and Methods.

**Chemicals and reagents**. BFA (Sigma Aldrich), ML141 (Tocris Bioscience) and LG186 (see synthesis section below) dissolved in dimethyl sulfoxide (DMSO) were used at 20 μg/ml, 10 μM and 10 μM respectively. ML141 and LG186 treatment was done for 30 min in serum free media and maintained during endocytic assays. Tetramethyl rhodamine-labeled dextran (TMR-Dex) (10,000 molecular weight; Molecular Probes, Thermofisher Scientific) was used at 1 mg/ml. The 4-hydroxy tamoxifen (Sigma Aldrich) was used at 3 μM to remove Dynamin 1/2/3 from the conditional dynamin triple knockout MEF cells as reported previously[28]. TrypLE express (GIBCO, Invitrogen) was used to detach cells according to the manufacturer's instruction. FuGENE HD transfection reagent (Promega) was used for transfection as per the manufacturer's instruction unless otherwise mentioned. SYLGARD 184 silicone elastomer kit (Dow Corning) was used to make PDMS sheets according to the manufacturer's instruction. For the reservoir experiments, cells were transfected with a membrane targeting plasmid pEYFP-mem (Clontech) using the Neon transfection device according to the manufacturer's protocol as described earlier[6].

**Endocytic and recycling assays**. CG endocytosis was monitored using fluorescent-dextran (TMR-Dextran) at 1 mg/ml in medium or fluorescent folate analog ($N^\alpha$-pteroyl-$N^\varepsilon$-Bodipy$^{TMR}$-L-lysine (PLB$^{TMR}$)) in folate free medium for the indicated time points at 37 °C. Endocytosis of TfR was monitored using 10 μg/ml fluorescent transferrin (Tf) at 37 °C incubation for indicated time points. Endocytosis was stopped using ice-cold HEPES-based buffer (M1) (M1:140 mM NaCl, 20 mM HEPES, 1 mM CaCl$_2$, 1 mM MgCl$_2$, 5 mM KCl, pH 7.4). To remove surface fluorescence, cells were treated with PI-PLC (50 μg/ml, 1 h; GPI-APs) or with ascorbate buffer (160 mM sodium ascorbate, 40 mM ascorbic acid, 1 mM MgCl$_2$, 1 mM CaCl$_2$, pH 4.5; Tf) at 4 °C, surface transferrin receptor labeled using anti-hTfR(OKT9) monoclonal antibody (used at 1:100) and subsequently fixed with 4% paraformaldehyde for 10 min.

To study endocytosis on de-adhering, cells were detached using TrypLE containing fluorescent-dextran at 1 mg/ml concentration for 3 min and the detached cells were pipetted into an ice-cold vial containing M1 buffer to stop the endocytosis. Cells were then re-plated back on coverslip bottom dish maintained at 4 °C, fixed, washed and imaged. To look at endocytosis in suspension, the cells soon after detaching were pipetted into a vial containing fluorescent-dextran kept at 37 °C. The volume was adjusted to have a final concentration of 1 mg/ml of the TMR-Dex and after 3 min the endocytosis was stopped by shifting vial to ice. The cells are spun down at 4 °C and then re-plated on coverslip bottom dish coated with ConA, maintained at 4 °C, fixed, washed and imaged.

To understand recycling of cargo on de-adhering, cells were pulsed with F-Dex for 3 min, quickly washed with M1 buffer at room temperature and then detached with TrypLE at 37 °C for 5 min, pipetted into a vial containing ice-cold M1 buffer and kept on ice. Cells were then re-plated back on coverslip bottom dish coated with ConA maintained at 4 °C, fixed, washed and imaged.

Endocytosis of fluid and transferrin at different temperatures were done by pre-equilibrating the cells to the respective temperatures and then pulsing TMR-Dex (fluid) or Tf-A647 (transferrin) for 5 min at these temperatures.

AP2 shRNA or its control pSUPER vector[37] was transfected into FR-AGS or HeLa cells for 4 days to knock down AP2- μ2 and inhibit CME as reported earlier[37] and was followed by endocytosis experiment as described above. Different small-molecule inhibitors were incubated with cells for 30 min in serum free media in their respective final concentrations and then medium was removed and pulsed with F-Dex at 1 mg/ml in serum free media containing the inhibitors since the inhibitor activity is reversible. Endocytosis was stopped by washing with ice-cold M1 buffer, fixed and imaged.

**Preparation of PDMS membrane ring**. Sylgard 184 silicone elastomer kit comes in two parts which are added in 10 to 1 mix ratio between the polydimethylsiloxane base and the curing agent. This is thoroughly mixed and degassing is done either using a vacuum desiccator or by centrifugation. To prepare polydimethylsiloxan (PDMS) sheets, 7 ml of this mixture was added to the middle of a circular 6 inch plate which is spun at 500 R.P.M for 1 min on a spin coater. This was cured at 65 °C overnight and then carefully peeled off either after treatment with oxygen plasma cleaner for 40 s or without treatment. This PDMS sheet is spread evenly and tightly placed between rings of the stretcher (Fig. 1b). The cells were plated in the middle of the PDMS sheet surrounded with water-soaked tissue paper to retain humidity and prevent drying up of medium. These rings were placed in the stretcher and

stretched by varying the level of vacuum as needed according to the calibration for the experiments.

**Stretch and osmolarity experiments**. For the stretch–relax experiments, cells plated on PDMS membrane were loaded on the cell stretcher system (Fig. 1b) within a temperature-controlled chamber at 37 °C. Vacuum was applied beneath the ring containing the PDMS sheet, deforming the membrane and stretching the cells plated on the PDMS. The setup was calibrated to stretch cells equi-biaxially to cause 6% strain for 90 s. Cells were pulsed for 90 s either during stretch or on releasing the strain. Medium containing F-Dex was kept at 37 °C in a water bath, and used for the endocytic pulse for the indicated time during or after stretch (see endocytic protocol above). Control cells were treated in the same way except for application of stretch.

For the osmolarity experiments, cells were treated with 50% hypotonic medium (unless otherwise mentioned) made with deionized water at 37 °C for the indicated time points and then pulsed with TMR-Dex either in hypotonic or isotonic medium as needed. The shock was applied for 60 s and pulse done for 60 s unless otherwise mentioned. For the extreme hypotonic experiment, 6× diluted medium was used for 10 min followed by isotonic pulse for 10 min to replicate the earlier protocol[39]. Endocytosis was stopped with ice-cold M1 buffer, washed, fixed and imaged.

**Optical tweezer measurements**. Tether forces were measured using a custom built optical tweezer using IR laser (continuous wave,1064 nm, TEM_00,1 W) along with 100×, 1.3 NA oil objective and motorized stage on an Olympus IX71 inverted microscope. Uncoated polystyrene beads added to the imaging chamber were allowed to settle and then held in the optical trap while simultaneously imaging through bright field on a coolsnap HQ CCD camera. Uncoated polystyrene beads bind to the membrane due to non-specific interactions. Membrane tethers are formed by attaching the beads to the cell membrane for a few seconds and by moving the bead away from the cell using the piezo stage. Uncoated polystyrene beads bind to the membrane due to non-specific interactions and a thin membrane tether is formed from the cell to the bead (Fig. 5a). The tether is held at a constant length and the fluctuation in the trapped bead is detected by using a quadrant photodiode which in turn is acquired and saved using a LabVIEW program through a Data Acquisition Card (USB-6009 NI). The trap stiffness is calibrated using the power spectrum method. The displacement of the bead from the center along with the trap stiffness is used to calculate the tether forces live using a custom written LabVIEW code.

**CTxB-HRP uptake and DAB reaction and Electron Microscopy**. WT and Cav$^{-/-}$ MEFs were de-adhered at room temperature followed by internalization of 4 μg/ml CTxB-HRP (Invitrogen) at 37 °C for 5 min, washed two times with ice-cold phosphate-buffered saline (PBS) followed by incubation on ice for 10 min with 1 mg/ml 3,3′-diaminobenzidine (DAB; Sigma Aldrich) with 50 μM ascorbic acid. This is followed by a 10-min treatment with DAB, ascorbic acid and 0.012% $H_2O_2$ and then washed twice with ice-cold PBS. Cells were fixed using 2.5% Glutaraldehyde (ProSciTech) at room temperature for 1 h followed by PBS wash for two times and then washed with 0.1 M Na cacodylate and left in the same for overnight at 4 °C. Cells were contrasted with 1% osmium tetroxide and 4% uranyl acetate. Cells were dehydrated in successive washes of 70%, 90% and 100% ethanol before embedding using 100% LX-112 resin at 60 °C overnight. Sections were viewed under a transmission electron microscope (JEOL 1011; JEOL Ltd. Tokyo, Japan), and electron micrographs were captured with a digital camera (Morada; Olympus) using AnalySIS software (Olympus).

**Imaging and analysis and statistics**. The quantification of endocytic uptake for a population is done by imaging on 20×, 0.75 NA on a Nikon TE300 wide-field inverted microscope. For the stretch experiments, an upright microscope (Nikon eclipse Ni–U) was used with a water immersion objective (60×, 1.0 NA). For endosome size calculation, spinning disk confocal microscope (100×, 1.4 NA) was used with ANDOR iQ software followed by analysis using 3D object counter plugin in Fiji[61]. The images were analyzed using MetaMorph® or Micro-Manager software and were processed for presentation using Adobe Illustrator. All images displayed are equally scaled for intensity unless otherwise mentioned. The integrated intensities, spread area and thus average uptake per cell were determined by drawing regions around each cell using the region measurement option in Fiji[61]. For plotting endocytic uptake, all values are normalized to the mean value of the control and plotted as a box plot using Origin software (OriginLab, Northampton, MA). Box plot shows uptake per cell (each data point) and it also shows median (middle line), standard deviation (whiskers), 25th percentile (lower line of box) and 75th percentile (upper line of box) value. The box plot contains points pooled from two separate experiments with technical duplicates in each and normalized to their respective controls. The total number of cells in each condition (pooled from all experiments) is mentioned in the legends. Statistical significance was tested using the Mann–Whitney test and p values used to determine significance are reported in the legends. The scale bar is 10 μm, unless otherwise mentioned.

## Data availability
Data supporting the findings of this manuscript are available from the corresponding author upon reasonable request.

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

## Acknowledgements

We thank Pietro De Camilli (Yale University, USA) for conditional Dynamin triple knockout cell line, Daniel Rösel (Charles University, Prague) for vinculin-null cell line, Darius V. Köster for the caveolin-null cell line, David J. Stephens (University of Bristol, UK) for an initial gift of LG186, Philippe Benaroch (Institut Curie, Paris) for AP2 shRNA, Clare M. Waterman (NIH, USA) for vinculin constructs, Feroz M.H. Musthafa (CCAMP, Bangalore) and G.V. Soni (RRI, Bangalore) for help with preparation of PDMS membrane. We would like to thank Manoj Mathew and central imaging and flow cytometry facility (CIFF, NCBS) for help with imaging, Dev Kumar (Mech. Workshop) for making components for stretch–relax apparatus and imaging, Dr. Anusuya Banerjee for help with illustrations, K. Joseph Mathew for final cartoon and thank members of P.P., X.T. and P.R-C. laboratories for hosting and helping J.J.T. with day-to-day experiments. X.T. acknowledges support from the Spanish Ministry of Economy and Competitiveness (BFU2015-65074-P), the Generalitat de Catalunya (2014-SGR-927) and the European Research Council (ERC-2013-CoG-616480). This work was supported by the Spanish Ministry of Economy and Competitiveness (BFU2016-79916-P to P.R.-C.), the European Commission (H2020-FETPROACT-01-2016-731957 to X.T. and P.R.-C.) and Obra Social 'La Caixa'. A.E.-A. acknowledges support by Juan de la Cierva Fellowship from Spanish Ministry of Economy and Competitiveness (IJCI-2014-19156). This study was also supported by grants SAF2014-51876-R from Spanish Ministry of Economy and Competitiveness (MINECO) and co-funded by FEDER funds to M.A.d.P., and 674/C/2013 from Fundació La Marató de TV3 to P.R.-C. and M.A.d.P. R.G.P. was supported by the National Health and Medical Research Council (NHMRC) of Australia (program grant, APP1037320 and Senior Principal Research Fellowship, 569452), and the Australian Research Council Centre of Excellence (CE140100036). We acknowledge the Australian Microscopy & Microanalysis Research Facility at the Center for Microscopy and Microanalysis at The University of Queensland. J.J.T. acknowledges pre-doctoral fellowship from Council for Scientific and Industrial Research (CSIR), Government of India. S.M. would like to acknowledge J.C. Bose Fellowship from DST, Government of India, and Wellcome Trust-DBT Margdarshi fellowship (IA/M/15/1/502018).

## Author contributions

J.J.T. and S.M. conceived the study. J.J.T., A.J.K., A.E.-A., P.P., P.R-C., M.A.d.P, R.G.P. and S.M. designed the experiments. Experiments were done and analyzed by J.J.T. (all experiments except reservoir resorption and CD44 uptake), A.J.K. (reservoir formation and resorption) and N.C. (CD44 uptake). M.C.G. and R.G.P. performed the EM experiments designed by R.G.P. and M.A.d.P. S.P. and P.P. built the optical tweezer setup. X.T. and P.R-C. designed and built the stretch system. S.S., P.P.S. and R.V. synthesized LG186. A.K., A.S.V. and M.R. wrote the model. J.J.T. and S.M. wrote the paper with inputs from other authors.

## Additional information

**Competing interests:** The authors declare no competing interests.

