## [Peer Review File · Nature Communications]

Reviewers' comments:

Reviewer #1 (Remarks to the Author):

This study investigates how endocytosis is coupled to membrane tension variations induced by cell stretching or cell deadhesion. The data presented here convincingly indicate that among the different endocytic pathways tested, the CLIC/GEEC (CG) pathway plays a prominent role in rapidly retrieving the excess of plasma membrane that follows stress relaxation. They further identify vinculin as a possible regulator of this process. Endocytosis, by retrieving the excess of plasma membrane that follows stress relaxation, and exocytosis, by providing additional membrane to prevent membrane rupture under increased membrane tension, have long been known to play a key role in membrane tension homeostasis under various mechanical cues. While the work presented here represents a much more thorough study than previously published studies, the level of novelty resides in the identification of the CG pathway as a major contributor in this process. It is nevertheless unclear whether the findings represent the level of novelty required for publication in Nature Communication as a recent study published this year in Cell Reports, which should be cited and discussed in this manuscript, has already identified the CG pathway in HeLa and colon cancer cells as the main endocytic pathway that is activated following a decrease in membrane tension (Holst et al. 2017).

- The authors refer to VLD as the "vacuole like dilations" that are formed upon recovery from hypo-osmotic shock. It is unclear how the CG pathway is related to VLDs. Indeed VLDs are not pictured on the final model presented in Fig. 8. The authors imply that CG endocytosis is required for the resorption of VLDs. Are VLDs connected with CG endosomes? Is cargo that is endocytosed by the CG pathway delivered into VLDs? The resorption of VLDs is inhibited by CytoD treatment (Komalska, Nat Commun 2015), so what happens to the CG pathway under CytoD treatment? In this context, it is interesting that GRAF1, another identified regulator of the CG pathway, was recently shown to be recruited to forming VLDs (Holst Cell Rep 2017). It is therefore important to document the role of GRAF1 in the process studied here.

- Better information regarding the kinetics of this response should be provided as most experiments are performed at 90 sec post stress. Live cell imaging monitoring the kinetics of CLIC structure formation together with VLD resorption should be provided by following the uptake of CLIC cargoes such as GPI or CD44 after different times of stress relaxation.

- Membrane tension measurements based on membrane tether pulling is an important piece of information in this study. The measurements however are performed at steady state after up or down regulation of the CG pathway. To better link these measurements to the process studied here, tether forces should be measured during the stretch-relax procedure.

- Although the data presented here seem to exclude caveolae, it is still intriguing that they are not involved in this regulation. Several groups including authors of this study have shown that caveolae are disassembled upon increased membrane tension, a process that prevents membrane rupture and cell death. Conversely, it is expected that stress relaxation would favor caveolae endocytosis, in agreement with the study by del Pozo (ref 30 herein) where cell deadhesion results in increased caveolae endocytosis. EM showing the presence of caveolae at the plasma membrane (sup Fig 2d) does not allow to conclude that caveolae endocytosis is not involved. Caveolae endocytosis should be analyzed by live cell imaging as done before (see Boucrot JCS 2011). This is important as it was reported that the CG pathway can be regulated by caveolae (Parton PLoS Biology 2014). In this context, it is interesting that an interaction between caveolin and vinculin was recently reported by the Nabi group (MBoC 2017). Repeating the vinculin experiments in Cav1 KO MEF would bring more information as the authors mention that "the precise mechanism behind the ability of vinculin to regulate the CG pathway is not understood...."

Reviewer #2 (Remarks to the Author):

In this manuscript, the authors address a very important and still debated question about the origin of tension homeostasis in cells. It has already been established that cells use endo- and exocytosis to regulate their membrane area and respond to membrane tension changes, but which particular pathway is used and how the tension set-point is controlled is not so clear. Thottacherry et al use different methods to change cell membrane tension, practically by inducing a membrane tension decrease either by stretching cells and relax them abruptly or by de-adhering them and observing their relaxation to basal state. They convincingly demonstrate that the CLIC/GEEC (GC) pathway is directly involved in the rapid elimination of the membrane reservoirs induced during the process, at least in cells where this pathway is operative. During cell stretching, they also observe a decrease of the uptake. The dynamin-dependent (CME or CIE) or caveolin-dependent pathways seem not to be involved in this mode of tension regulation. Their experiments show that the tension set-point can be modulated by changing the level of key regulators of the GC pathway. Moreover, they have also interestingly identified vinculin as a member of the mechanosensing machinery that responds in a negative feedback loop for the tension regulation. I find that this paper opens new perspectives on the role of non-conventional endocytic pathways on the mechanical regulation of cell plasma membrane and should eventually be published in Nature Communications, providing that some points are clarified or complemented.

1- Vinculin is normally considered as part of the focal adhesion (FA) complex. But, talin is the mechanosensitive element : when a force opens it, its activation leads to vinculin binding, thus vinculin acts downstream of talin. Would it be possible to test whether talin is also involved in tension sensing? At this stage, the connection between vinculin, the CG pathway and the GEF GBF1 appears quite phenomenological. Is there a direct coupling between adhesion and tension regulation? What would be the molecular mechanism? I guess it will require much more work to establish this, but it could be at least discussed. Moreover, in the experiments, the methods to change membrane tension have at the same time a mechanical action on FA and eventually on vinculin, even when cells are detached since both adhesion and tension are released. Is there a way to make a hypertonic shock or aspirate a cell in suspension to check if in non-adhering conditions, vinculin is still involved in the mechanosensing machinery?

2. The authors show that CG regulates tension either by resorbing large membrane reservoirs when tension amplitude changes are large, but also in the case of sub-macroscopic deformations. It is not clear for me how the same molecular endocytic machinery practically works in these very different situations, and internalize a large range of membrane patch sizes. Could the authors comment on this point? Moreover, since CG is also used for controlling the level of some specific membrane proteins in the plasma membrane, how much tension control does interfere with protein traffic?

3. Figure 5d (magnetic beads to measure cell stiffness): the method is minimally detailed and the result and interpretation of the measures are not clear.

Reviewer #3 (Remarks to the Author):

Review of Mechanochemical feedback and control of endocytosis and membrane tension.

The manuscript by Thottacherry et al. entitled, 'Mechanochemical feedback and control of endocytosis and membrane tension', the authors present an experimental study on how membrane tension and CG pathway for endocytosis forms a mechanochemical feedback. This is an

important question in the field of trafficking and the authors have conducted a considerable number of experiments to investigate how membrane tension affects the CG pathway. There has been a considerable interest in recent times about the role of membrane tension and endocytosis. Here, the authors study a dynamin and clathrin-independent pathway, CG endocytosis and conclude that tension and CG act in a feedback loop. The main novelty of this study is the two-way analysis -- changing tension through multiple modes affects the extent of CG pathway and changing the extent of the CG pathway affects membrane tension. This is demonstrated quite convincingly through a multitude of experiments. I have the following comments for the authors to consider to strengthen their study.

Major comments:

- 1) Membrane tension -- in cells, membrane tension is closely related to the cortical actin organization. The authors should clarify what they mean by membrane tension and how one is to interpret this. They allude to this somewhat in line 258, but this can be discussed better and perhaps earlier.
- 2) Multiple modes of inducing tension: how do they affect the cytoskeleton? The authors used multiple modes of tension induction -- stretch relaxation, spreading/deadhering, osmotic stress -- to induce membrane tension. How do these different modes affect the underlying cortical cytoskeleton? How do we know that the only impact of these tension inducing modes is on the plasma membrane?
- 3) Role of tension in CME -- from my reading of the manuscript, it appears that the authors suggest that tension doesn't play an important role in regulating CME. See page 2, line 58 for example. But there are many references that indicate that CME is impacted by tension, only a few of which are given below. Can the authors expand on their discussion or at least clarify the message they are seeking to provide with respect to CME?

Bucher et al. 2017; Walani, Torres, and Agrawal 2015; Ferguson et al. 2017; Hassinger et al. 2017.

- 4) There are two separate parts to this study -- the feedback between CG and tension and the role of actin cytoskeleton proteins. The first part is a convincing study. However, the study on the role of vinculin appears to be only a preliminary investigation. It is not clear to me why vinculin alone was chosen for this study and how other focal adhesion proteins might play a role. I suggest that the authors conduct a separate, in-depth investigation of focal adhesion proteins rather than just adding one set of experiments here.
- 5) Use of different cell types -- Why did the authors use HeLa and CHO cells in certain experiments (Figure 5)? I was trying to follow the rationale for this and understand the arguments made in lines 280-285 but couldn't. The authors should clarify this.
- 6) A mechanistic, theoretical model that explains how membrane mechanics, tension, and the CG pathway interact could strengthen the paper. By this, I don't mean an in-depth computational study, but rather a physical model that can help develop intuition for how membrane properties, tension, and GBF1 interact that will put all the experimental observations in context.

Minor comments:

1. The authors should proofread the text carefully. There are some spelling mistakes and awkward sentences in the text throughout.
2. This is a matter of personal preference but the title has two ands and doesn't necessarily reflect the particular endocytic pathway being studied. Perhaps the authors can revise the title to be more specific?
3. Abbreviations appear in a number of places without being fully expanded upon.

Bucher, Delia, Felix Frey, Kem A. Sochacki, Susann Kummer, Jan-Philip Bergeest, William J. Godinez, Hans-Georg Kraeusslich, et al. 2017. "Flat-to-Curved Transition during Clathrin-Mediated Endocytosis Correlates with a Change in Clathrin-Adaptor Ratio and Is Regulated by Membrane Tension." bioRxiv, July. Cold Spring Harbor Labs Journals,

162024.

Ferguson, Joshua P., Scott D. Huber, Nathan M. Willy, Esra Aygün, Sevde Goker, Tugba Atabey, and Comert Kural. 2017. "Mechanoregulation of Clathrin-Mediated Endocytosis." *Journal of Cell Science* 130 (21):3631–36.

Hassinger, Julian E., George Oster, David G. Drubin, and Padmini Rangamani. 2017. "Design Principles for Robust Vesiculation in Clathrin-Mediated Endocytosis." *Proceedings of the National Academy of Sciences of the United States of America* 114 (7):E1118–27.

Walani, Nikhil, Jennifer Torres, and Ashutosh Agrawal. 2015. "Endocytic Proteins Drive Vesicle Growth via Instability in High Membrane Tension Environment." *Proceedings of the National Academy of Sciences* 112 (12). National Academy of Sciences: E1423–32.

Reviewer #4 (Remarks to the Author):

This paper reports evidence that a reduction of membrane tension triggers a fast and transient increase of endocytosis and fluid uptake. Tension variation is obtained by different means: stretch-release cycles, de-adhesion, or osmotic shock. Evidence is shown that the increased uptake upon membrane tension reduction is neither due to clathrin mediated endocytosis, nor is it dynamin dependent. The authors link the increased uptake to the so-called CLIC/GEEC endocytosis pathway, and identify a molecular mechanism for the mechano-sensitivity of through the involvement of vinculin.

In general, I find the paper quite sensible and interesting. The notion of a negative feedback between tension and endocytosis that could help maintain an homeostatic tension has been around for some time, but it is nice to see experimental evidence for it, and to have this process linked with a particular pathway, with hints of a molecular mechanism. I think this is a very useful paper.

My comments are two-fold. On the one hand, I think the authors are at places over-interpreting their data, and would like to see this amended. I also find that a more precise characterisation of the dynamical nature of the mechanochemical feedback would be very valuable. These comments are developed below.

1) L.177: "Together, these experiments indicated that the clathrin, dynamin or caveolin dependent endocytic mechanisms do not exhibit a rapid respond to a reduction in membrane tension." What is shown here is that there is a rapid response to membrane tension reduction that is clathrin, dynamin, and caveolin independent, but not that these pathways do not exhibit such response. This sentence should be changed to reflect the actual findings.

2) L.201 and below. The author claim that the tension-dependent endocytosis mechanism creates a feedback loop hat et membrane tension to an homeostatic value. Is tension regulation also possible for HeLa cells, which lack CG endocytosis? Could the author show the evolution of tether force on HeLa cell upon stretch-release, or hypo-osmotic shock. It is likely that these cells also show a transient variation of tension, which would suggest, at best, the existence of redundant tension regulation mechanisms. However, the dynamics of tension relaxation could be very different, perhaps not as fast as the one shown here to depend upon CG endocytosis.

3) L.255 and Supl. Fig. 5: Calling the response to a magnetic tweezer "membrane stiffness" seems completely misleading to me. If only the membrane were involved to that response, one would expect a stiffness in the range of tens of pN/ μm at the most (close to the value of membrane tension), while the one measured here is 1000 larger, and is certainly not solely due to the membrane mechanical response. While an effect can be seen upon LG186 treatment (Fig.S5 d-f), I

find it very doubtful that this can be directly related to variation of membrane tension, as stated in the text. Consequently, the sentence (L: 257): " That this effect was due to a reduction in membrane tension and not any effects on the cytoskeleton, was corroborated..." is, to my view, inconsistent with numerical estimates and not acceptable. Short of a clear understanding of what the so-called "membrane stiffness" measured by magnetic tweezers actually means, and how it is related to membrane mechanics, this sentence should be either heavily watered down or removed, together with the magnetic trap experiments.

4) The tether force is used to quantify membrane tension at steady-state under different conditions. I imagine that tether force measurement is done on adhered cells, although I don't think this is explicitly mentioned (it should be). As a number of different factors may affect the steady-state tension, tether force measurement should also be performed in the mechanically perturbed state (stretched, or hypo-osmotic shock), under different condition to test the mechanical feedback hypothesis.

It is presumably possible to measure the temporal evolution of the tether force during the transient response of cell to perturbation, especially in the case of stretch-release cycles. Could these experiments be performed to see if there is a direct correlation between the evolution of the tether force and reservoir resorption (as in Fig.4).

The dynamics of recruitment of the CG machinery (Fig.6) could also be monitored dynamically before, during, and after the osmotic shock. This could also be combined with optical tweezers measurement of tether force to obtain a direct dynamical correlation between the two processes. Such measurement would greatly support the claim of mechanical feedback and its involvement in setting the homeostatic tension.

5) Fig.4. Why is there no quantification for $T=37$ deg.?

Reviewer #5 (Remarks to the Author):

In this paper, the authors reported that the CLIC/GEEC (CG) pathway of endocytic process is regulated by membrane tension in a vinculin dependent manner. The results are interesting and the evidences are compelling. I would like to recommend publication of this work after the authors clarify a few points and make changes accordingly.

One of the key experiments is the use of optical tweezers to measure the force in membrane tethers. The author didn't provide information on how the bead is attached to the membrane. Did they use ConA-coated bead similar to what used in their magnetic tweezers experiment? Further details of how this membrane tether was formed are needed.

In the optical tweezers experiments, the changes in the tether force were used to establish a link between the CG pathway and the membrane tension. However, the tether force depends on other conditions such as the length of the tether, the locations on the cells where the measurement is done, and membrane adhesion to cytoskeleton. The authors need to clarify how these conditions are controlled such that the level of tether forces they measured can be correlated with the endocytic process through the CG pathway.

The authors also investigated the correlation between the local stiffness of the cell and the endocytic process through the CG pathway. The local stiffness was measured by applying 0.5 nN force pulses using a magnetic tweezers device to ConA-coated magnetic beads that were attached to the cell membrane and detecting the resulting bead displacement. This force is one order of magnitude greater than the tether force measured using the optical tweezers. I wonder why such a large force didn't produce a protruding member tether. Does it imply a strong membrane attachment to cytoskeleton that prevents formation of the membrane tether? If this is the case,

then this membrane stiffness measurement does not reflect membrane tension, and cannot be compared with the tether force measured using optical tweezers. The implications of the results from the magnetic tweezers measurement on the endocytic process through the CG pathway need to be re-discussed.

The authors stated "That this effect was due to a reduction in membrane tension and not any effects on the cytoskeleton, was corroborated by the lack of a change in the measured stiffness of fibronectin-coated beads attached to cells via integrin-fibronectin..." Based on this sentence, it seems that the authors suggest that the local stiffness measured by the magnetic tweezers device is not due to membrane adhesion to the cytoskeleton. If this is what the authors implied, then they need to quantitatively establish a link between the local membrane stiffness they measured and the membrane tension, and need to explain why the protruding membrane tethers did not form in the magnetic tweezers experiments.

The authors identified vinculin as a key player in the observed mutual dependence between the CG pathway of endocytic process and membrane tension, but they didn't provide a clue of how vinculin might be involved in this regulation. Although it would be too much to ask the authors to completely elucidate the mechanism of the involvement of vinculin in this regulation, it will be very helpful if they provide information whether it requires vinculin activation through binding to talin at focal adhesion. Repeating the measurement in talin-deleted cells can provide this piece of information.

Point by point response to **reviewers' comments:**

Please note all Figure numbers from Main and Supplementary refer to the new
Figures.

**Reviewer #1 (Remarks to the Author):**

**This study investigates how endocytosis is coupled to membrane tension**
**variations induced by cell stretching or cell deadhering. The data presented**
**here convincingly indicate that among the different endocytic pathways**
**tested, the CLIC/GEEC (CG) pathway plays a prominent role in rapidly**
**retrieving the excess of plasma membrane that follows stress relaxation.**
**They further identify vinculin as a possible regulator of this process.**
**Endocytosis, by retrieving the excess of plasma membrane that follows**
**stress relaxation, and exocytosis, by providing additional membrane to**
**prevent membrane rupture under increased membrane tension, have long**
**been known to play a key role in membrane tension homeostasis under**
**various mechanical cues. While the work presented here represents a much**
**more thorough study than previously published studies, the level of novelty**
**resides in the identification of the CG pathway as a major contributor in this**
**process. It is nevertheless unclear whether the findings represent the level**
**of novelty required for publication in Nature Communication as a recent**
**study published this year in Cell Reports, which should be cited and**
**discussed in this manuscript, has already identified the CG pathway in HeLa**
**and colon cancer cells as the main endocytic pathway that is activated**
**following a decrease in membrane tension (Holst et al. 2017).**

We thank this reviewer for his/her critical review of the manuscript and for
appreciating a role for the CG pathway in tension response. There are three parts
to the significance of our finding. First, we extensively test all the major endocytic
pathways and identify CG endocytosis as the major pathway that acutely responds
to changes in tension. Second, we measure tether forces to explore the role of
endocytic pathways on membrane tension. Finally, we find that this pathway is
under a mechano-chemical control, by identifying a key mechanotransducer,
vinculin, in this process.

In the revised version we further strengthen our understanding of the molecular
mechanism of this regulation with additional experiments with knock out lines
that fail to exhibit CG endocytosis, various vinculin constructs, and relevant
molecular perturbations. Finally, we provide a theoretical framework to integrate
these observations.

With regard to Holst et al 2017¹, we would like to point out that we had cited and
discussed this manuscript in our original submission (Ref. 47/Line no394/Page
10). Holst et al, was published while our manuscript was under review elsewhere,
and couple of months before we put up our manuscript on 'bioRxiv' and
submission for review at Nature communications. We believe that Holst et al, is
looking at a qualitatively different response. Holst et al, suggests that a GRAF1
mediated endocytic mechanism which occurs in HeLa and certain colon cancer

cells suppress blebbing. This is relevant since blebbing-mediated migration may
be important for metastasis. They conclude that a clathrin-independent pathway
may be involved in this response, since GRAF1 knock down affects the hypotonic-
isotonic (Hypo-Iso) stimulated uptake whereas AP2- knock down is less effective.

In this regard, we note that Holst et al differs from our study in two main aspects.
Firstly, this study uses HeLa cells which have been shown to have a predominantly
clathrin-mediated endocytic pathway² (further tested in our study and elaborated
below), and secondly the conditions for the tension-decrease stimulated endocytic
process is very different from the conditions we have used in our study. We
address these issues below to clarify that we are indeed looking at a very different
process from that explored by Holst et al 2017.

1) On knocking down AP2, fluid-phase uptake is completely shut down in the HeLa
cell line used in the Holst et al study (see Fig 1D:AP2 siRNA-Isotonic in Holst et al,
2017). This is consistent with an earlier study that conclusively show CME is the
predominant endocytic pathway in this cell line². We further confirm this
observation using AP2 shRNA and find that indeed fluid-phase uptake is
predominantly clathrin (AP2)-mediated in these cells (Supplementary Fig 5a).
Consistent with a shutting down of the clathrin-mediated pathway in HeLa cells;
the AP2 shRNA causes a decrease in transferrin (Tf) uptake, and an increase in
surface transferrin receptor (TfR). We have tested Tf uptake, TfR surface levels
and fluid-phase uptake in the same cells to avoid variability and this is quantified
following three color imaging. However, in cells that have a functioning CG
pathway, the same AP2 shRNA mediated knockdown while inhibiting Tf uptake
and enhancing TfR surface levels, results in an increase in fluid-phase uptake
(Supplementary Fig 5b). This increase in fluid uptake upon AP2 knock down is
similar to inhibiting dynamin-mediated CME³ (Figure 5c). In fact, this uptake takes
place almost entirely by the CG pathway, since it is almost completely inhibited on
inhibiting the CG pathway (Figure 5c). The CG pathway takes in the major fraction
of extracellular fluid along with other cargoes as reported earlier while Tf that
traffics through CME does not colocalize with fluid in these cells. This pathway is
regulated by CDC42, ARF1, GBF1 and IRSP53 and is independent of dynamin, AP2
or Clathrin⁴⁻⁷. Further, HeLa cells do not have a robust CG pathway⁸. We find that
HeLa cells do not respond to inhibition of GBF1 (Supplementary Fig 5c). GBF1, a
GEF for ARF1, also not seem to be present in an active form at the cell surface in
HeLa cells unlike CHO with bonafide CG pathway (Supplementary Fig 5d/5e).
Thus HeLa cells do not have a robust non-CME pathway.

2) To understand the endocytic process contributing to the increase in fluid
uptake in Holst et al, we repeated the Holst protocol in HeLa cells and in a cell line
that completely lacks the CG pathway. We have recently shown that IRSP53-/-
null cell lines do not have a functional CG pathway, and re-addition of IRSP53
restores this pathway⁷. It should be noted that the Holst protocol is different from
our Hypo-Iso protocol; we used an application of a 50% hypotonic solution for one
minute prior to restoration of isotonicity for the same length of time. On the other
hand, and Holst et al uses either water or 6x water diluted isotonic solution for 10
minutes, preceding a return to isotonicity. Using the Holst protocol, we see a
similar enhancement of fluid uptake in all cell lines we have examined, whereas

using the protocol that we have developed to modulate membrane tension, either
stretch-relax, or hypo-iso, is enhanced only in cells that exhibit a bonafide CG
endocytic pathway (Supplementary Fig 7a and 7b).

In the Holst protocol which represents a drastic change in osmolarity, we find that
many cells are distorted and dextran-filled structures are located outside phase
contrast areas demarcating cells (Arrows and Insets, Supplementary Fig 7a and
7b). This suggests that in many instances the membrane may have been
completely separated from the cortex, and a massive endocytic engulfment may
have initiated as a result. Thus, we conclude that the mechanism being assessed
by Holst is very different from the responses we have addressed. It is also likely
that such a mechanism may also preferentially utilize GRAF1, but does not appear
to utilize the CG machinery.

Therefore, we suggest that it is likely that cells may utilize different mechanisms
depending on the extent of the hypotonic shock, and the resultant change in
membrane tension. Importantly, under the conditions of modulating membrane
tension as described in our studies, the CG pathway is the dominant mechanism
for the restoration of membrane tension and area. Blebbing appears to be a
cellular response to the inhibition of all endocytic mechanisms⁹ and Holst et al
2017 seem to have uncovered the importance of this in the migration of cancer
cells. In the manuscript we discuss the significance of the Holst et al. study, and
propose an alternative explanation for their results (see page no 6, line no 217 and
discussion- page 11, line 473).

We have extensively characterized the role of different endocytic pathways using
multiple means of modulating tension and more importantly we measure tether
forces to quantify membrane tension. Our study proposes to understand the role
of these endocytic processes in responding to changes in membrane tension. Our
work also brings out a novel link between the focal adhesion protein, vinculin, in
regulating the CG pathway by a mechano-chemical feedback mechanism. This is a
hitherto unappreciated connection between mechanical inputs and chemistry. We
are confident that the readers of *Nature Communications* would find the
mechanochemical control of an endocytic pathway and its importance in
maintaining membrane homeostasis, not only noteworthy, but of broad
significance.

- **The authors refer to VLD as the “vacuole like dilations” that are formed**
**upon recovery from hypo-osmotic shock. It is unclear how the CG pathway is**
**related to VLDs. Indeed VLDs are not pictured on the final model presented**
**in Fig. 8. The authors imply that CG endocytosis is required for the**
**resorption of VLDs. Are VLDs connected with CG endosomes? Is cargo that is**
**endocytosed by the CG pathway delivered into VLDs? The resorption of VLDs**
**is inhibited by CytoD treatment (Komalska, Nat Commun 2015), so what**
**happens to the CG pathway under CytoD treatment? In this context, it is**
**interesting that GRAF1, another identified regulator of the CG pathway, was**
**recently shown to be recruited to forming VLDs (Holst Cell Rep 2017). It is**
**therefore important to document the role of GRAF1 in the process studied**
**here.**

We have used multiple means to change membrane tension, namely 'stretch-
relax', deadhering, and osmotic shocks to study morphological changes in the cell
membrane and endocytic responses ensuing. 'Reservoirs' and 'VLDs' represent
similar types of membrane responses that differ in shape (see our previous
manuscript; Kosmalka et al., 2015), formed by stretch-relax and hypotonic-
isotonic shift, respectively¹⁰. Subsequent to both these responses we observe a
very similar CG endocytic response. The final model in the original manuscript
tries to assimilate these multiple responses to change in tension and hence VLDs
are not specifically depicted in our figure. (Note that in the revised manuscript,
the final model is modified to reflect the theoretical model that integrates our
findings of endocytic response and membrane tension.) To be clear, VLDs are a
fast 'passive' membrane invagination in response to changes in osmolarity that is
followed by 'active' endocytic response to take in the excess membrane,
accumulated to accommodate the hypotonic treatment. Since VLDs are passive
membrane invaginations at the cell surface, cargo endocytosed by CG cannot be
delivered to VLDs. Please refer to the reviewer's figure that explains this
(Reviewer Fig 1). Our experiments indicate that formation of this passive
structure is not necessary for the ensuing endocytic response. By growing cells on
a hydrogel, the water that is pumped out of the cell during the restoration of
osmolarity, is resorbed by the hydrogel, preventing the generation of VLDs as we
observed previously¹⁰ (Supplementary Fig 9b). However, the excess membrane
continues to be retrieved via upregulation of CG pathway even in the absence of
VLD formation (Supplementary Fig 9c). We thank the reviewer for pointing this
out and hope that the rewritten version helps the readers as well.

As previously mentioned, the role of GRAF1 has mostly been studied in HeLa cells
that lack the CG pathway. In Holst et al, GRAF1 levels (either on inducing or
knocking down the same) appear to correspondingly modulate AP2 levels that
could explain its effect on fluid uptake since fluid uptake is AP2 dependent in HeLa
cells (Holst et al., Fig 1E)¹. GRAF1 associates with CME derived endosomes² and
CG-derived endosomes and affect the maturation (and probably recycling) of
endosomes¹¹. GRAF1 also appears to be recruited to the plasma membrane and
VLDs during the extreme osmotic shock protocol in Holst et al. that is independent
of CG pathway. The exact role of GRAF1 in CG and CME pathway would require an
in-depth study of its own which is not the focus of this manuscript. Here we
propose a role for CG endocytic pathway in mediating an acute response to
moderate and possibly more physiological membrane tension changes. It is
important to note that we propose that a constitutive endocytic pathway that is
dynamin, clathrin independent, but dependent on CDC42, ARF1 (and its GEF,
GBF1), as well as IRSp53, is upregulated to respond to acute but moderate
alterations in membrane tension, mediated by different mechanisms. Whilst in the
Holst protocol, cells that do not exhibit this form of CG endocytic mechanism
(HeLa cells and IRSp53 null cells) also exhibit a similar endocytic response to the
extreme change in osmolarity.

*Reviewer Figure 1: The cells on shifting to isotonic from hypotonic medium (Hypo –*
*Iso) shows formation of VLD¹⁰. Further, there is an increase in endocytosis following*
*which the membrane morphology is restored.*

- **Better information regarding the kinetics of this response should be**
**provided as most experiments are performed at 90 sec post stress. Live cell**
**imaging monitoring the kinetics of CLIC structure formation together with**
**VLD resorption should be provided by following the uptake of CLIC cargoes**
**such as GPI or CD44 after different times of stress relaxation.**

We would like to clarify that the fluid-phase uptake response that we measure
occurs contemporaneously with strain relaxation. Our assay for dextran uptake is
for 90 seconds to integrate the endocytic uptake over this time-period, even in the
control cells at steady state. Importantly, we find that the increase in endocytosis
is a fast transient event that is lost in 90 seconds (Fig 1c; Stretch-Relax-Wait).
Further, this also coincides with the time it takes to remove the membrane
reservoirs (Compare Fig 1c with Fig 4b). Attempts at imaging these two fast
processes have been difficult at present, and a microfluidic chamber to carry out
rapid pH buffer change using SEC-GFP-GPI^{7,12} while changing tension would be
necessary for this purpose. While these experiments are currently planned, it
requires the development of new instrumentation in our laboratories and is
currently beyond the scope of this manuscript.

- **Membrane tension measurements based on membrane tether pulling is an**
**important piece of information in this study. The measurements however**
**are performed at steady state after up or down regulation of the CG pathway.**
**To better link these measurements to the process studied here, tether forces**
**should be measured during the stretch-relax procedure.**

We measure steady state tension of a cell to understand the importance of
endocytic process since we see that CG pathway can respond to changes in tension.
Cells at steady state would be maintained at a specific membrane tension due to
the simultaneous functioning of a number of endo-exocytic processes, adhesion.
Thus it is necessary for us to look at the importance of modulating the endocytic
process on the steady state tension, as measured by monitoring tether pulling
force.

However, we agree with the reviewer's suggestion of monitoring tension of a cell
during the stretch-relax and other procedures, and we are developing the
instrumentation to do the same. Currently, the PDMS-based stretch-relax device
is incompatible with the optical tweezer set up, due to the inherently scattering
nature of the PDMS on which the cells have to be grown.

The case of osmotic shock is simpler but still poses problems as a sudden change
of medium causes large enough flow disturbance to detach the bead from the
tether. We need to improve the techniques to enable the measurements the
referee has suggested and feel that these experiments are also beyond the scope
of this current manuscript.

**- Although the data presented here seem to exclude caveolae, it is still**
**intriguing that they are not involved in this regulation. Several groups**
**including authors of this study have shown that caveolae are disassembled**
**upon increased membrane tension, a process that prevents membrane**
**rupture and cell death. Conversely, it is expected that stress relaxation**
**would favor caveolae endocytosis, in agreement with the study by del Pozo**
**(ref 30 herein) where cell deadhering results in increased caveolae**
**endocytosis. EM showing the presence of caveolae at the plasma membrane**
**(sup Fig 2d) does not allow to conclude that caveolae endocytosis is not**
**involved. Caveolae endocytosis should be analyzed by live cell imaging as**
**done before (see Boucrot JCS 2011). This is important as it was reported that**
**the CG pathway can be regulated by caveolae (Parton PLoS Biology 2014). In**
**this context, it is interesting that an interaction between caveolin and**
**vinculin was recently reported by the Nabi group (MBoC 2017). Repeating**
**the vinculin experiments in Cav1 KO MEF would bring more information as**
**the authors mention that "the precise mechanism behind the ability of**
**vinculin to regulate the CG pathway is not understood....**

The data from caveolin null cells which exhibit a normal transient upregulation of
the CG pathway, indeed excludes the role of caveolae/caveolin in this transient
process. More pertinently, the results from the dynamin Triple Knock Out not only
exclude a role for clathrin-mediated endocytosis in this response, it also confirms
the lack of any role for caveolae this fast active retrieval of plasma membrane (Fig
3a), since endocytic uptake via caveolae is dynamin-dependent¹³.

We have tried to discuss the roles of different endocytic pathways in the
discussion to highlight this (see page 11, line 460). By forming membrane
invaginations caveolae help counter increase in tension by passively releasing
extra membrane and preventing rupture in a passive manner¹⁴. Regarding the
increase in caveolar endocytosis on deadhering¹⁵, it is to be noted that the
caveolar uptake appears to be triggered once the cells are placed in suspension
and this continues for several minutes and up to couple of hours, taking in
membrane of specific composition, containing GM1 and cholesterol. The kinetics
are therefore very different to the CG pathway response shown here, as discussed
in the revised manuscript. The caveolar pathway may act as a timer for the time
the cell is in suspension helping to control anoikis¹⁵, and may function in parallel
to this fast transient endocytic response. On the contrary, the CG pathway is

acutely upregulated only on reducing tension by stretch-relaxation process or
during deadhering but swiftly returns to steady state once the excess membrane
is removed (Fig. 1C and 1D). However, we find that if cells are left in suspension
for 10 minutes, the fluid uptake reduces in contrast to the increase in endocytosis
expected from cargo that is internalized via the caveolar pathway (Supplementary
Fig. 10d). Thus, various endocytic pathways with different time scales appear to
operate, potentially for a variety of reasons.

**Reviewer #2 (Remarks to the Author):**

**In this manuscript, the authors address a very important and still debated**
**question about the origin of tension homeostasis in cells. It has already been**
**established that cells use endo- and exocytosis to regulate their membrane**
**area and respond to membrane tension changes, but which particular**
**pathway is used and how the tension set-point is controlled is not so clear.**
**Thottacherry et al use different methods to change cell membrane tension,**
**practically by inducing a membrane tension decrease either by stretching**
**cells and relax them abruptly or by de-adhering them and observing their**
**relaxation to basal state. They convincingly demonstrate that the CLIC/GEEC**
**(GC) pathway is directly involved in the rapid elimination of the membrane**
**reservoirs induced during the process, at least in cells where this pathway**
**is operative. During cell stretching, they also observe a decrease of the**
**uptake. The dynamin-dependent (CME or CIE) or caveolin-dependent**
**pathways seem not to be involved in this mode of tension regulation. Their**
**experiments show that the tension set-point can be modulated by changing**
**the level of key regulators of the GC pathway. Moreover, they have also**
**interestingly identified vinculin as a member of the mechanosensing**
**machinery that responds in a negative feedback loop for the tension**
**regulation. I find that this paper opens new perspectives on the role of non-**
**conventional endocytic pathways on the mechanical regulation of cell**
**plasma membrane and should eventually be published in Nature**
**Communications, providing that some points are clarified or complemented.**

We thank the reviewer for her/his support of our work. We are pleased that our
data was able to convince this reviewer of the importance of the CG pathway and
its relationship to a mechanosensing machinery. We have done further
experiments to complement the present study and clarified the points below.

**1- Vinculin is normally considered as part of the focal adhesion (FA)**
**complex. But, talin is the mechanosensitive element : when a force opens it,**
**its activation leads to vinculin binding, thus vinculin acts downstream of**
**talin. Would it be possible to test whether talin is also involved in tension**
**sensing? At this stage, the connection between vinculin, the CG pathway and**
**the GEF GBF1 appears quite phenomenological. Is there a direct coupling**
**between adhesion and tension regulation? What would be the molecular**
**mechanism? I guess it will require much more work to establish this, but it**
**could be at least discussed. Moreover, in the experiments, the methods to**
**change membrane tension have at the same time a mechanical action on FA**

**and eventually on vinculin, even when cells are detached since both**
**adhesion and tension are released. Is there a way to make a hypertonic**
**shock or aspirate a cell in suspension to check if in non-adhering conditions,**
**vinculin is still involved in the mechanosensing machinery?**

We thank the reviewer for the set of constructive comments. We have now tested
the role of talin, and focal adhesions in general, for regulating this process.

Vinculin along with Talin is part of the mechanosensitive module of the focal
adhesion¹⁶ and is known to be important for mechanotransduction¹⁷. An obvious
experiment to test a role for Talin requires depleting both talin 1 and 2.
Unfortunately, this causes cells to round up in culture¹⁸ making it difficult to do
any endocytic experiments that involves multiple washes. Therefore, we took an
alternative approach and have tested different vinculin mutants (Vin-CA:
Constitutively active and does not require Talin to activate it, VinA50I: Talin
binding mutant, Vin-A50I-CA: Constitutively active and cannot bind talin) in the
Vinculin $-/-$ back ground¹⁹. We find that Vin-CA mutant inhibit CG pathway and
does not show a response to decrease in tension unlike WT (Fig. 7c/7e). Vin-CA
that cannot bind to Talin (Vin-A50I-CA) shows a similar phenotype (Fig. 7e),
however a wild type Vin which is unable to bind Talin and therefore be activated
by Talin is not able to restore the sensitivity to the change in membrane tension
(Fig. 7d). This indicates that the talin is required to activate vinculin in response
to changes in tension while active vinculin is required to negatively regulate the
CG endocytic pathway (Also see Page 9, Line 358).

We also looked at cells in suspension to further dissect the endocytic response to
changes in tension. Cells in suspension lack any focal adhesions and interestingly,
showed no increase in endocytic uptake on hypo-iso shift unlike the attached cells
(Supplementary Fig. 10d). These experiments indicate the focal adhesion
signaling in general and talin in particular are important for activating vinculin to
show this transient endocytic response. We have now expanded the discussion to
reflect the possible ways vinculin could regulate the CG pathway through GBF1.

**2. The authors show that CG regulates tension either by resorbing large**
**membrane reservoirs when tension amplitude changes are large, but also in**
**the case of sub-macroscopic deformations. It is not clear for me how the**
**same molecular endocytic machinery practically works in these very**
**different situations, and internalize a large range of membrane patch sizes.**
**Could the authors comment on this point? Moreover, since CG is also used**
**for controlling the level of some specific membrane proteins in the plasma**
**membrane, how much tension control does interfere with protein traffic?**

This reviewer has raised an interesting point. The early endosomes in the CG
pathway seem to be pleomorphic²⁰ and makes vesicles of different sizes. This is
likely to internalize differential amount of material depending on the conditions.
We have now done high resolution imaging to understand the size and number of
endosomes during an osmotic release. We find that the number of endosomes as
well as the size increases (Supplementary Fig. 3b/3c/3d, Movie 1 and 2).

Since the CG pathway has machinery to substantially upregulate its endocytic
capacity, it is consistent with the increase in uptake on changing tension. CG
pathway is only transiently upregulated when the tension is reduced and comes
back to the steady state (Fig. 1C, 1D). The increase in size of endosome points to a
mechanism that could regulate the timing of scission of endosome to control size
of an endosome in relation to the tension. However, further study is required to
understand the mechanism. There is a substantial amount of recycling from the
CG pathway and it could help in sorting the contents of the endosome²⁰. It is likely
that this sorting event would help restore the change in membrane composition if
any. However, the lipid composition regulation through this high capacity
pathway is an open question and we are pursuing this.

**3. Figure 5d (magnetic beads to measure cell stiffness): the method in**
**minimally detailed and the result and interpretation of the measures are not**
**clear.**

On advice from reviewer 4, (see our detailed explanation below- Reviewer 4,
Comment 3) we have now removed the experiments utilizing the magnetic
tweezer in this revised manuscript.

**Reviewer#3(Remarks to the Author):**

**Review of Mechanochemical feedback and control of endocytosis and**
**membrane tension.**

**The manuscript by Thotacherry et al. entitled, ‘Mechanochemical feedback**
**and control of endocytosis and membrane tension’, the authors present an**
**experimental study on how membrane tension and CG pathway for**
**endocytosis forms a mechanochemical feedback. This is an important**
**question in the field of trafficking and the authors have conducted a**
**considerable number of experiments to investigate how membrane tension**
**affects the CG pathway. There has been a considerable interest in recent**
**times about the role of membrane tension and endocytosis. Here, the**
**authors study a dynamin and clathrin-independent pathway, CG**
**endocytosis and conclude that tension and CG act in a feedback loop. The**
**main novelty of this study is the two-way analysis -- changing tension**
**through multiple modes affects the extent of CG pathway and changing the**
**extent of the CG pathway affects membrane tension. This is demonstrated**
**quite convincingly through a multitude of experiments.**

We thank the reviewer for her/his encouraging comments and pointing out the
novelty of this study and appreciate the comments made to strengthen our
manuscript.

**I have the following comments for the authors to consider to strengthen**
**their study.**

Major comments:

**1) Membrane tension -- in cells, membrane tension is closely related to the**
**cortical actin organization. The authors should clarify what they mean by**
**membrane tension and how one is to interpret this. They allude to this**
**somewhat in line 258, but this can be discussed better and perhaps earlier.**

This is an important point and we have now revised this to bring it earlier in the
main text. (See page 7, line number 271).

**2) Multiple modes of inducing tension: how do they affect the cytoskeleton?**
**The authors used multiple modes of tension induction -- stretch relaxation,**
**spreading/deadhering, osmotic stress -- to induce membrane tension. How**
**do these different modes affect the underlying cortical cytoskeleton? How**
**do we know that the only impact of these tension inducing modes in on the**
**plasma membrane?**

The plasma membrane is intricately linked with the cortical cytoskeleton through
ERM proteins, focal adhesion molecules and cell – cell adhesion etc. It would be
impossible to decouple these effects individually. The term apparent membrane
tension here involves all these factors and we have now rewritten the manuscript
to better reflect these to the reader (See page 7, line number 271). Further, we
compare tether forces only between conditions where we change CG pathway and
show that if the CG pathway is absent, such effects on tether forces are absent as
well (Fig. 5f).

**3) Role of tension in CME -- from my reading of the manuscript, it appears**
**that the authors suggest that tension doesn't play an important role in**
**regulating CME. See page 2, line 58 for example. But there are many**
**references that indicate that CME is impacted by tension, only a few of**
**which are given below. Can the authors expand on their discussion or at**
**least clarify the message they are seeking to provide with respect to CME?**
**Bucher et al. 2017; Walani, Torres, and Agrawal 2015; Ferguson et al. 2017;**
**Hassinger et al. 2017.**

This is an important point and we have made changes in the discussion to reflect
this (see page 11, line 462). Further we have included these references to make
the discussion more comprehensive.

**4) There are two separate parts to this study -- the feedback between CG and**
**tension and the role of actin cytoskeleton proteins. The first part is a**
**convincing study. However, the study on the role of vinculin appears to be**
**only a preliminary investigation. It is not clear to me why vinculin alone was**
**chosen for this study and how other focal adhesion proteins might play a**
**role. I suggest that the authors conduct a separate, in-depth investigation of**
**focal adhesion proteins rather than just adding one set of experiments here.**

We thank this reviewer for her/his constructive comment. We have explained this
in reply to Reviewer number 2 in the first major comment.

**5) Use of different cell types -- Why did the authors use HeLa and CHO cells**
**in certain experiments (Figure 5)? I was trying to follow the rationale for this**
**and understand the arguments made in lines 280-285 but couldn't. The**
**authors should clarify this.**

We have now rewritten the manuscript to better reflect the argument regarding
the use of different cell lines (Page 5, line 196 and Page 7, Line 300). For the
purpose of this study, HeLa is used as a negative control since we observe that
these cells do not have a constitutive CG pathway.

**6) A mechanistic, theoretical model that explains how membrane**
**mechanics, tension, and the CG pathway interact could strengthen the paper.**
**By this, I don't mean an in-depth computational study, but rather a physical**
**model that can help develop intuition for how membrane properties,**
**tension, and GBF1 interact that will put all the experimental observations in**
**context.**

We have now incorporated a physical model that can help develop intuition for
how membrane properties and GBF1 interact and this allows us to put all the
experiments into one framework. See page 10, line 390 and Supplementary
Section for details.

**Minor comments:**
**1. The authors should proofread the text carefully. There are some spelling**
**mistakes and awkward sentences in the text throughout.**

We have proof read the text and have asked an outside lab colleague to help
improve the readability of the manuscript.

**2. This is a matter of personal preference but the title has two ands and**
**doesn't necessarily reflect the particular endocytic pathway being studied.**
**Perhaps the authors can revise the title to be more specific?**

The title is now changed to increase the specificity and remove repeating words
as follows "Mechanochemical feedback control of dynamin independent
endocytosis modulates membrane tension in adherent cells".

**3. Abbreviations appear in a number of places without being fully expanded**
**upon.**

We have now made sure that the abbreviations are listed and fully expanded on
being used at the first time. Hope this reviewer comment helps make it easier for
the readers.

**Bucher, Delia, Felix Frey, Kem A. Sochacki, Susann Kummer, Jan-Philip**
**Bergeest, William J. Godinez, Hans-Georg Kraeusslich, et al. 2017. "Flat-to-**
**Curved Transition during**
**Clathrin-Mediated Endocytosis Correlates with a Change in Clathrin-**
**Adaptor Ratio and Is Regulated by Membrane Tension." bioRxiv, July. Cold**
**Spring Harbor Labs Journals,**

162024.

Ferguson, Joshua P., Scott D. Huber, Nathan M. Willy, Esra Aygün, Sevde
Goker, Tugba Atabay, and Comert Kural. 2017. "Mechanoregulation of
Clathrin-Mediated Endocytosis."
Journal of Cell Science 130 (21):3631-36.

Hassinger, Julian E., George Oster, David G. Drubin, and Padmini
Rangamani. 2017. "Design Principles for Robust Vesiculation in Clathrin-
Mediated Endocytosis." Proceedings of the National Academy of Sciences of
the United States of America 114 (7):E1118-27.

Walani, Nikhil, Jennifer Torres, and Ashutosh Agrawal. 2015. "Endocytic
Proteins Drive Vesicle Growth via Instability in High Membrane Tension
Environment." Proceedings of the
National Academy of Sciences 112 (12). National Academy of
Sciences:E1423-32.

Reviewer #4 (Remarks to the Author):

This paper reports evidence that a reduction of membrane tension triggers
a fast and transient increase of endocytosis and fluid uptake. Tension
variation is obtained by different means: stretch-release cycles, de-
adhesion, or osmotic shock. Evidence is shown that the increased uptake
upon membrane tension reduction is neither due to clathrin mediated
endocytosis, nor is it dynamin dependent. The authors link the increased
uptake to the so-called CLIC/GEEC endocytosis pathway, and identify a
molecular mechanism for the mechano-sensitivity of through the
involvement of vinculin. In general, I find the paper quite sensible and
interesting. The notion of a negative feedback between tension and
endocytosis that could help maintain an homeostatic tension has been
around for some time, but it is nice to see experimental evidence for it, and
to have this process linked with a particular pathway, with hints of a
molecular mechanism. I think this is a very useful paper.

We thank the reviewer for his/her encouraging remarks on the manuscript.

My comments are two-fold. On the one hand, I think the authors are at places
over-interpreting their data, and would like to see this amended. I also find
that a more precise characterisation of the dynamical nature of the
mechanochemical feedback would be very valuable. These comments are
developed below.

1) L.177: "Together, these experiments indicated that the clathrin, dynamin
or caveolin dependent endocytic mechanisms do not exhibit a rapid respond
to a reduction in membrane tension."What is shown here is that there is a
rapid response to membrane tension reduction that is clathrin, dynamin,

**and caveolin independent, but not that these pathways do not exhibit such**
**response. This sentence should be changed to reflect the actual findings.**

We have now changed the wordings accordingly in the main text of the manuscript
(see page 5, line 171).

**2) L.201 and below. The author claim that the tension-dependent**
**endocytosis mechanism creates a feedback loop hat et membrane tension to**
**an homeostatic value. Is tension regulation also possible for HeLa cells,**
**which lack CG endocytosis? Could the author show the evolution of tether**
**force on HeLa cell upon stretch-release, or hypo-osmotic shock. It is likely**
**that these cells also show a transient variation of tension, which would**
**suggest, at best, the existence of redundant tension regulation mechanisms.**
**However, the dynamics of tension relaxation could be very different,**
**perhaps not as fast as the one shown here to depend upon CG endocytosis.**

We agree with the reviewer on this. There could be other mechanisms at work
here using cortical cytoskeleton etc. It is also possible that the exocytic
mechanisms that we have not probed here could have a role in maintaining
homeostasis. We find that HeLa cells do respond to extreme perturbations that is
independent of CG pathway (Supplementary Fig. 7a, 7b). Even vinculin null cells
respond to higher changes in tension (Fig. 7b). These observations indicate that
there could be multiple mechanisms responding to changes in tension. We are
currently trying to create a set up to measure tension live while being able to
modulate the strain but these experiments are beyond the scope of the current
manuscript.

**3) L.255 and Supl. Fig. 5: Calling the response to a magnetic tweezer**
**“membrane stiffness” seems completely misleading to me. If only the**
**membrane were involved to that response, one would expect a stiffness in**
**the range of tens of pN/μm at the most (close to the value of membrane**
**tension), while the one measured here is 1000 larger, and is certainly not**
**solely due to the membrane mechanical response. While an effect can be**
**seen upon LG186 treatment (Fig.S5 d-f), I find it very doubtful that this can**
**be directly related to variation of membrane tension, as stated in the text.**
**Consequently, the sentence (L:257):” That this effect was due to a reduction**
**in membrane tension and not any effects on the cytoskeleton, was**
**corroborated...” is, to my view, inconsistent with numerical estimates and**
**not acceptable. Short of a clear understanding of what the so-called**
**“membrane stiffness” measured by magnetic tweezers actually means, and**
**how it is related to membrane mechanics, this sentence should be either**
**heavily watered down or removed, together with the magnetic trap**
**experiments.**

In magnetic tweezers experiments, concanavalin A-coated beads are attached to
the cell membrane via glycoproteins, and were then pulled. Since the membrane
is linked to the cytoskeleton, we fully agree with the reviewer that the
measurement reflects both membrane and cytoskeletal properties. We note that
the experiments are not directly comparable to optical tweezers measurements

for two reasons. First, the beads are bigger (4 μm) and attached to cells via a large
part of their surface, increasing mechanical resistance. Second and most
important, magnetic tweezers exert force largely in the plane of the membrane,
whereas optical tweezers experiment pulls membrane tethers perpendicular to
the membrane plane. Cell membrane bilayers may be extended in the
perpendicular direction, they are likely to be extremely stiff in-plane, due to the
effects of the membrane and the cytoskeleton interaction. As the reviewer notes,
we performed experiments by directly linking beads to the cytoskeleton through
fibronectin and integrins, which abolishes the measured differences. In our view,
the difference we observe are related to alterations in membrane tension, and the
discrepancy with tether values is due to the very different experimental setups.
However, we agree with the reviewer that this is not definitive proof, and
therefore we conducted optical tweezers experiments.
In the interest of preventing any confusion, we have taken the advice of the
reviewer and have removed the magnetic tweezer data. We also provide a better
explanation with regards to the membrane tether measurements (see page 7, line
274).

**4) The tether force is used to quantify membrane tension at steady-state**
**under different conditions. I imagine that tether force measurement is done**
**on adhered cells, although I don't think this is explicitly mentioned (it should**
**be). As a number of different factors may affect the steady-state tension,**
**tether force measurement should also be performed in the mechanically**
**perturbed state (stretched, or hypo-osmotic shock), under different**
**condition to test the mechanical feedback hypothesis. It is presumably**
**possible to measure the temporal evolution of the tether force during the**
**transient response of cell to perturbation, especially in the case of stretch-**
**release cycles. Could these experiments be performed to see if there is a**
**direct correlation between the evolution of the tether force and reservoir**
**resorption (as in Fig.4). The dynamics of recruitment of the CG machinery**
**(Fig.6) could also be monitored dynamically before, during, and after the**
**osmotic shock. This could also be combined with optical tweezers**
**measurement of tether force to obtain a direct dynamical correlation**
**between the two processes. Such measurement would greatly support the**
**claim of mechanical feedback and its involvement in setting the homeostatic**
**tension.**

Yes, the tether forces are measured on adherent cells to understand the effect of
modulating endocytic pathways on steady state tension. We have now modified
the text to convey this message more clearly (Page 7, Line 272). The cells are
grown on a PDMS substrate for the stretch-relax experiments. Unfortunately,
PDMS scatters the laser light preventing accurate measurement of tension, in the
configuration that we make the measurements. Therefore, it is currently not
possible to measure tension while cells are grown on PDMS, and we work in an
inverted microscope configuration. We understand that measuring tether forces
while making changes to tension in a live cell is important and we are developing
the right tools to measure tension changes while making other perturbations. We
hope to discuss these results and more in future manuscripts.

**5) Fig.4. Why is there no quantification for T=37 deg.?**

The values were normalized to the T= 37C. To avoid confusion, we have now
replotted the figure to show the quantification and normalization clearly.

**Reviewer #5 (Remarks to the Author):**

**In this paper, the authors reported that the CLIC/GEEC (CG) pathway of**
**endocytic process is regulated by membrane tension in a vinculin**
**dependent manner. The results are interesting and the evidences are**
**compelling. I would like to recommend publication of this work after the**
**authors clarify a few points and make changes accordingly.**

We thank the reviewer for his/her positive remarks and recommendation for
publication. Please find the point by point response below.

**One of the key experiments is the use of optical tweezers to measure the**
**force in membrane tethers. The author didn't provide information on how**
**the bead is attached to the membrane. Did they use ConA-coated bead**
**similar to what used in their magnetic tweezers experiment? Further details**
**of how this membrane tether was formed are needed.**

We have now added a more detailed explanation of the experiment (see page 7,
line 272 and Methods – Page 23, Line 948).

We have used ConA coated beads in magnetic tweezer and also uncoated
polystyrene beads that binds to the plasma membrane due to non-specific
interactions in optical tweezer. However, similar beads are used between control
and test conditions wherever it has been used and these have been mentioned in
the Methods.

**In the optical tweezers experiments, the changes in the tether force were**
**used to establish a link between the CG pathway and the membrane tension.**
**However, the tether force depends on other conditions such as the length of**
**the tether, the locations on the cells where the measurement is done, and**
**membrane adhesion to cytoskeleton. The authors need to clarify how these**
**conditions are controlled such that the level of tether forces they measured**
**can be correlated with the endocytic process through the CG pathway.**

In cells the steady-state tether force, as measured after the initial force during
formation has relaxed, is independent of the tether length. This is presumably
because of the tension regulation mechanisms operating in the cells. So one is not
working at constant lipid number. This has been studied previously^{21,22}. We do not
know if the tension is going to be different at different locations. However, we do
not use polarized cells in our study that could have different tension at different
location (apical vs basal, leading edge vs trailing edge etc.). We consistently pull
tethers from lamellipodia and average our data over several tethers. This should

average out such dependencies. Spatial dependency is of interest in its own right.
It has been shown previously that the steady state tether force depends on the
adhesion to the actin cortex. The force is slightly reduced when the cortex is
removed. This has been attributed to membrane-cortex adhesion. In our
experiments we are only making relative comparisons of tether force between
cells with active CG pathway and those with this pathway inhibited. As opposed to
absolute measurement of membrane in-plane tension, the cortex effects are
expected to be negligible in such membrane tether measurements. We have now
clarified all three points in the main text.

**The authors also investigated the correlation between the local stiffness of**
**the cell and the endocytic process through the CG pathway. The local**
**stiffness was measured by applying 0.5 nN force pulses using a magnetic**
**tweezers device to ConA-coated magnetic beads that were attached to the**
**cell membrane and detecting the resulting bead displacement. This force is**
**one order of magnitude greater than the tether force measured using the**
**optical tweezers. I wonder why such a large force didn't produce a**
**protruding member tether. Does it imply a strong membrane attachment to**
**cytoskeleton that prevents formation of the membrane tether? If this is the**
**case, then this membrane stiffness measurement does not reflect membrane**
**tension, and cannot be compared with the tether force measured using**
**optical tweezers. The implications of the results from the magnetic tweezers**
**measurement on the endocytic process through the CG pathway need to be**
**re-discussed**

**The authors stated "That this effect was due to a reduction in membrane**
**tension and not any effects on the cytoskeleton, was corroborated by the**
**lack of a change in the measured stiffness of fibronectin-coated beads**
**attached to cells via integrin-fibronectin..." Based on this sentence, it seems**
**that the authors suggest that the local stiffness measured by the magnetic**
**tweezers device is not due to membrane adhesion to the cytoskeleton. If**
**this is what the authors implied, then they need to quantitatively establish**
**a link between the local membrane stiffness they measured and the**
**membrane tension, and need to explain why the protruding membrane**
**tethers did not form in the magnetic tweezers experiments.**

In the interest of preventing any confusion, we have taken the advice of this and
reviewer 4 (response 3) and have removed the magnetic tweezer data.

**The authors identified vinculin as a key player in the observed mutual**
**dependence between the CG pathway of endocytic process and membrane**
**tension, but they didn't provide a clue of how vinculin might be involved in**
**this regulation. Although it would be too much to ask the authors to**
**completely elucidate the mechanism of the involvement of vinculin in this**
**regulation, it will be very helpful if they provide information whether it**
**requires vinculin activation through binding to talin at focal adhesion.**
**Repeating the measurement in talin-deleted cells can provide this piece of**
**information.**

This is an important point raised by this reviewer and reviewer 2, and has been
clarified above (see point 1; reviewer 2).

REFERENCES

1. Holst, M. R. *et al.* Clathrin-Independent Endocytosis Suppresses Cancer Cell
Blebbing and Invasion. *Cell Rep.* **20**, 1893–1905 (2017).

2. Bitsikas, V., Corrêa, I. R. & Nichols, B. J. Clathrin-independent pathways do
not contribute significantly to endocytic flux. *Elife* **3**, e03970 (2014).

3. Park, R. J. *et al.* Dynamin triple knockout cells reveal off target effects of
commonly used dynamin inhibitors. *J. Cell Sci.* **126**, 5305–12 (2013).

4. Sabharanjak, S., Sharma, P., Parton, R. G. & Mayor, S. GPI-anchored proteins
are delivered to recycling endosomes via a distinct cdc42-regulated,
clathrin-independent pinocytic pathway. *Dev. Cell* **2**, 411–23 (2002).

5. Kumari, S. & Mayor, S. ARF1 is directly involved in dynamin-independent
endocytosis. *Nat. Cell Biol.* **10**, 30–41 (2008).

6. Gupta, G. D. *et al.* Analysis of endocytic pathways in Drosophila cells
reveals a conserved role for GBF1 in internalization via GEECs. *PLoS One* **4**,
e6768 (2009).

7. Sathe, M. *et al.* Small GTPases and BAR domain proteins regulate branched
actin polymerisation for clathrin and dynamin-independent endocytosis.
*Nat. Commun.* **9**, 1835 (2018).

8. Kalia, M. *et al.* Arf6-independent GPI-anchored protein-enriched early
endosomal compartments fuse with sorting endosomes via a
Rab5/phosphatidylinositol-3'-kinase-dependent machinery. *Mol. Biol. Cell*
**17**, 3689–704 (2006).

9. Norman, L. L. *et al.* Cell blebbing and membrane area homeostasis in
spreading and retracting cells. *Biophys. J.* **99**, 1726–33 (2010).

10. Kosmalska, A. J. *et al.* Physical principles of membrane remodelling during
cell mechanoadaptation. *Nat. Commun.* **6**, 7292 (2015).

11. Francis, M. K. *et al.* Endocytic membrane turnover at the leading edge is
driven by a transient interaction between Cdc42 and GRAF1. *J. Cell Sci.*
4183–4195 (2015). doi:10.1242/jcs.174417

12. Taylor, M. J., Perrais, D. & Merrifield, C. J. A high precision survey of the
molecular dynamics of mammalian clathrin-mediated endocytosis. *PLoS*
*Biol.* **9**, (2011).

13. Henley, J. R., Krueger, E. W. A., Oswald, B. J. & McNiven, M. A. Dynamin-
mediated internalization of caveolae. *J. Cell Biol.* **141**, 85–99 (1998).

14. Sinha, B. *et al.* Cells respond to mechanical stress by rapid disassembly of
caveolae. *Cell* **144**, 402–13 (2011).

15. del Pozo, M. a *et al.* Phospho-caveolin-1 mediates integrin-regulated
membrane domain internalization. *Nat. Cell Biol.* **7**, 901–8 (2005).

16. Stutchbury, B., Atherton, P., Tsang, R., Wang, D.-Y. & Ballestrem, C. Distinct
focal adhesion protein modules control different aspects of
mechanotransduction. *J. Cell Sci.* **130**, 1612–1624 (2017).

17. Atherton, P., Stutchbury, B., Jethwa, D. & Ballestrem, C. Mechanosensitive
components of integrin adhesions: Role of vinculin. *Exp. Cell Res.* **343**, 21–
27 (2016).

18. Zhang, X. *et al.* Talin depletion reveals independence of initial cell

spreading from integrin activation and traction. *Nat. Cell Biol.* **10**, 1062–
1068 (2008).

19. Case, L. B. *et al.* Molecular mechanism of vinculin activation and nanoscale
spatial organization in focal adhesions. *Nat. Cell Biol.* **17**, 880–892 (2015).

20. Howes, M. T. *et al.* Clathrin-independent carriers form a high capacity
endocytic sorting system at the leading edge of migrating cells. *J. Cell Biol.*
**190**, 675–91 (2010).

21. Datar, A., Bornschlögl, T., Bassereau, P., Prost, J. & Pullarkat, P. A. Dynamics
of membrane tethers reveal novel aspects of cytoskeleton-membrane
interactions in axons. *Biophys. J.* **108**, 489–497 (2015).

22. Dai, J. & Sheetz, M. P. Cell membrane mechanics. *Methods Cell Biol.* **55**,
157–171 (1998).

Reviewers' comments:

Reviewer #1 (Remarks to the Author):

With respect to the Holst paper, I partially disagree with the interpretation made by the authors. While this is correct that GRAF1 depletion increased cell blebbing, it was also shown that a decrease in surface tension was buffered by clathrin-independent endocytosis, a process regulated by GRAF1. Whether this is a peculiarity of HeLa cells is not so clear as the role of clathrin-independent endocytosis in buffering membrane tension decrease was also established in colon cancer cells in the same study. We obviously disagree on this point and this is why the simplest answer to this was to test GRAF1.

While it may be indeed difficult to adapt the PDMS-based stretch-relax device to nanotube pulling, several groups have used hypo-iso osmotic cycles to measure the dynamics of membrane tension variations. I disagree therefore that these experiments are beyond the scope of this study as the coupling between endocytosis and membrane tension variations is intrinsically dynamic and is key to the processes investigated here.

Reviewer #2 (Remarks to the Author):

I think that the authors did a great job answering my comments as well as those of my colleagues. The new data on the role of Talin and of the activation of Vinculin are convincing. I also like very much the mechanical/chemical feedback model. It helps understanding how the tension set point could be regulated. There is still work to do to refine the model and understand how to relate time scales to the actual experiments, but I think that at this stage the paper can be published

Reviewer #3 (Remarks to the Author):

In the revised version of this manuscript, the authors have addressed all of my previous comments satisfactorily. It would have been easier to re-review if the authors would have marked up the changes to the text in a different color.

Reviewer #4 (Remarks to the Author):

The authors have taken into account some of my comments regarding over-interpreting data, and have argued that the dynamical assessment of tether force that I proposed is not feasible at the moment. I am willing to accept this response. However, upon re-reading the manuscript and the response to reviewer 2 (question 1), it appears to me that there could be a major problem with the interpretation of the data, which I unfortunately overlooked upon my first reading, and which casts serious doubts on the design of the feedback loop shown in Fig.8.

The new results linking vinculin activation to Talin, and showing that cells in suspension do not show an increase in endocytosis upon hypo-iso shift are to my mind quite significant. So far, the discussion of this paper relies on a mechano-chemical feedback loop between membrane tension and endocytosis rate. I tend to think that the absence of such feedback in suspended cell suggest that the feedback does directly involve membrane tension, but rather the tension on Focal Adhesion (FA) elements in adherent cells. While the two types of tension could be related, and can both be decreased during stretch-release, de-adhesion, or hypo-iso shift experiments, it is not clear at all that there is a direct correlation between FA tension and membrane tension. This suggests that the feedback model (Fig.8b) is missing a key element, which is the tension on the adhesion sites, and which is related in an unclear and potentially complex way to membrane

tension in adherent cells. As far as I am concerned, this represents a major difference from the direct feedback loop proposed in this paper. Indeed, one could imagine treatment that would affect the tension on adhesion sites (such as inhibition of myosin contractility) could also lead to up-regulation of the CG endocytotic pathway without relying on obvious variation of the membrane tension itself.

A crucial control in that regard, would be to measure the changes in membrane tension of cells in suspension during a hypo-iso shift. I would expect that membrane tension does indeed decrease in such condition, without triggering the feedback on endocytosis described in this paper, since we already know that such cells do not show the modulation of GC endocytosis. That would show that variation of membrane tension may not be what is measured in the feedback loop, thus invalidating the direct robust feedback proposed in Fig.8.

In fact, as far as I know, it is rather unclear to see at the molecular level how membrane tension, which is a force (per unit length) actin in the plane of the membrane, could modulate the force felt by Talin/Vinculin complexes, which presumably has a sizeable component perpendicular to the membrane.

These considerations makes me hesitant to recommend publication without discussing the effect of cell adhesion and of FA tension. The fact that the title was amended to specifically restrict the study to adherent cells is good. The paper surely shows a correlation between reduction of membrane tension and increase of CG endocytosis, and this finding is worth publication. However, the causality aspect which is essential to talk about feedback loop has, in my opinion, not been clearly demonstrated. I am conscient of the fact that a discussion has been added on the "real" meaning of tension, which is now called an "apparent tension" that also involves membrane-cytoskeleton adhesion (l.277). I would like to stress that my reservations are of a different nature, and cannot be satisfied by simply stating that Talin/Vinculin tension also contributes to the apparent, or effective tension discussed in this article.

As I realise that changing the focus of the paper to FA tension might be too much to ask, I thus propose that the paper be amended so that all implication of causality between a decrease of membrane tension and an increase of endocytosis are removed, and replaced by a discussion of the existence of a clear correlation between the two quantities. For instance (the list is not exhaustive)

l.74: "we find that a clathrin, caveolin and dynamin-independent endocytic mechanism, the CLIC/GEEC (CG) pathway, rapidly responds to changes in membrane tension, acting to restore it to a specific set point."

should thus be modified to reflect the actual finding, for instance:

we find that a clathrin, caveolin and dynamin-independent endocytic mechanism, the CLIC/GEEC (CG) pathway, correlates with changes in membrane tension, and might be involved in setting a specific membrane tension set point."

or

l. 171: "171 Together, these experiments indicated that there is a rapid endocytic response to reduction in membrane tension that is clathrin-, dynamin-, and caveolin-independent."

should thus be modified, for instance:

there is a rapid endocytic response that correlates with a reduction in membrane tension, but relies on the activation of proteins involved in cell adhesion.

I think this is a crucial point.

minor comment:

- Check the sign in Eq.5 and 8 of the Supplementary Material.

Reviewer #5 (Remarks to the Author):

I have read through the revised manuscript and the replies. The authors have addressed most of my concerns from my previous review.

Point by point response to **reviewer's comments:**

(Note that all the revisions and text mentioned here have also been highlighted in the main text)

1) Reviewer #1 (Remarks to the Author):

With respect to the Holst paper, I partially disagree with the interpretation made by the authors. While this is correct that GRAF1 depletion increased cell blebbing, it was also shown that a decrease in surface tension was buffered by clathrin-independent endocytosis, a process regulated by GRAF1. Whether this is a peculiarity of HeLa cells is not so clear as the role of clathrin-independent endocytosis in buffering membrane tension decrease was also established in colon cancer cells in the same study. We obviously disagree on this point and this is why the simplest answer to this was to test GRAF1.

We thank the reviewer for her/his comments on our revised manuscript.

However, we wish to indicate (and reiterate) important points that this reviewer has failed to appreciate.

1) The protocols used for the change in osmolarity in Holst et al., and our manuscript are drastically different. While the protocol utilized by Holst creates a drastic change in osmolarity (either just water addition or >6x dilution of isotonic medium) for 10 min prior to reverting to isotonicity (Hypo-6x-Iso), the protocol adopted for the bulk of our work, is a 1 minute hypotonic (~ 50 % Isotonic) treatment followed by isotonicity (Hypo-iso). Upregulation of the CG endocytic activity under Hypo-Iso conditions is similar to what we see in cell detachment, and cell stretch-relax conditions (Supplementary Fig. 1c, Supplementary Fig. 6a, Fig. 1b, and Fig. 1c). On the other hand, cell lines that do not exhibit the CG pathway (HeLa Cells or the IRSp53 KOs), remain responsive only to the Hypo-6x-iso condition (Supplementary Fig. 7a and 7b). This indicates that the response being monitored in Holst et al, is functioning via a different mechanism, which may be GRAF1 and potentially AP2 dependent.

Osmotic changes could have multiple effects¹, which is why we have taken pains to establish three separate methods to test the role of tension on endocytic processes. Holst et al either uses just water or 6x-Hypo medium for 10 minutes to alter membrane properties. This causes dramatic morphological changes which persist even after a recovery to isotonic conditions (Supplementary Fig. 7a and 7b) and is clearly not the same process that we are studying at more moderate changes in osmolarity (Supplementary Fig. 1c).

2) This reviewer also fails to distinguish between clathrin-independent endocytosis as a generic catch all phrase for all endocytic pathways that function in the absence of clathrin, and a specific

clathrin-independent endocytic pathway, namely the CLIC/GEEC (CG) pathway that functions in the absence of both clathrin and dynamin.

A confusion that exists in categorizing various endocytic pathways is also reflected by this reviewer's response to our statements about the role of GRAF1. From available literature, GRAF1 has been implicated in endocytic pathways that function in Hela cells²⁻⁵. These cells lack a ARF1/CDC42 regulated CG pathway as shown by us previously⁶, (and this study) and also more extensively in Bitsikas et al⁷. Thus, perturbing GRAF1 is likely to have more widespread effects, and could affect more than one mechanism for endocytosis. The confusion arises when this reviewer equates the CG pathway with any GRAF1-sensitive clathrin-independent (or dependent) pathway.

The CG pathway whose role we are addressing in the control of membrane tension, requires the function of GBF1, ARF1, CDC42 and now the recently shown, IRSp53, but importantly not dynamin⁸⁻¹². While it may also utilize GRAF1, for the reasons cited above, the perturbation of GRAF1 and monitoring its effect on fluid phase endocytosis will not provide any further clarification of the endocytic mechanism influencing the Hypo-Iso treatment condition utilized by us.

3) Thus, we would like to reiterate (as we have in our previous response to this reviewer), since GRAF1 participates in both clathrin-dependent⁷ and clathrin-independent pathways², ascertaining whether GRAF1 plays a role in the CG pathway that responds to specific alterations in membrane tension, is not the goal of this study. We have other much clearer means to identify the endocytic mechanism that shows us that the CG pathway is a central player in this response.

a) we have tested the roles of 3 key players (GBF1, CDC42 and IRSP53) to be necessary for the membrane tension modulated operation of the CG pathway in this manuscript.

b) by using a combination of specific null mutants of dynamin and caveolin, and experiments addressing the endocytic uptake of ligands trafficking through CME and CG pathway, we conclude that the CG pathway specifically responds to tension changes we have utilized.

c) We also uncover the mechano-chemical transducer of this response, providing evidence that Vinculin, a mechanotransducer, mediates this strain-sensitive response.

Therefore, we believe testing yet another molecule such as GRAF1 will provide no additional insight into the understanding of the mechanism. More importantly, this is not the focus of the paper or part of the concerns raised by any of the other 4 reviewers. Further, reviewer 1 expressed agreement with our conclusion that CG pathway is important for this response in the previous round. Thus, we are quite perplexed about this repeated demand made on us to prove the role of GRAF1 in our pathway.

4) We are frankly unsure of GRAF1's role in the CG pathway and testing its role in is a whole new project by itself and currently beyond the scope of the current manuscript. Our reasoning behind our lack of clarity about the role of GRAF1 is the following:

a) GRAF1 has been shown to act in clathrin mediated endocytosis (CME) in addition to the papers which implicate it in clathrin independent endocytosis (CIE)⁷.

b) A major and serious concern about all GRAF1 published papers²⁻⁵ that suggest its role in CIE is that key experiments have been done in cells that lack bonafide CG endocytosis (eg. HeLa cells as extensively detailed in Bitsikas et al⁷, previously observed by us⁶, and further tested in the current manuscript). Importantly, Holst et al⁵ also shows that the basal fluid uptake is almost completely dependent on AP2, a CME regulator, raising questions about the existence of a CIE mechanism in these cells.

5) Equally important is the technical feasibility of the GRAF1 depletion studies. Unless we develop new reagents, this is somewhat beyond our immediate scope for the following reasons: Most GRAF1 studies have appeared from Richard Lundmark group²⁻⁵ and there are no commercial antibodies that are known to work to test its depletion. However, we also note that most of the experiments in the Holst et al paper from Richard's laboratory are conducted in HeLa cells stably expressing GRAF1-GFP and its depletion is proxy for endogenous GRAF1 depletion. siRNA against GRAF1 is made-to-order and is not readily available. We have ordered a set a few months ago from the same source and are yet to receive the same. Finally, these experiments by Lundmark group have all been conducted in HeLa cells where there is no evidence for the functioning of the CG pathway (our manuscript and previous studies^{6,7}).

Due to the abovementioned concerns, we believe role of GRAF1 in different endocytic pathway(s) requires extensive and careful characterization in multiple cell lines and context. We plan to undertake this as a separate study to understand the role of GRAF1 in CME and CIE endocytosis in multiple cell lines. Our current MS looks at the response of a specific endocytic pathways to changes in membrane tension and vice versa, and in this context we think exploring the role of GRAF1 will not further the scientific understanding in any meaningful way.

6) We believe that we addressed Reviewer 1's concerns with battery of experiments detailed in our revised manuscript, which were perhaps overlooked (by this reviewer). We have successfully replicated some of the experiments that were performed in Holst et al paper, for instance experiments using AP2 shRNA, and osmotic shock. We then go on to then show that the conditions that we employ in our MS are quite different from what was used in the Holst et al protocols (Supplementary 6a, 7a and 7b). We also show that AP2 depletion that inhibits CME pathway inhibits fluid uptake in HeLa cells similar to Holst et al. in direct contrast to the increase in uptake observed in cell line with a robust CG pathway (see Supplementary Fig. 5a and 5b). This underlies the fact that we are talking about different physiological context in our MS and again are quite perplexed by the attempts of Reviewer 1 to club our work with Holst et al.

7) Minor comment about the reviewer's statement - "Whether this is a peculiarity of HeLa cells is not so clear as the role of clathrin-independent endocytosis in buffering membrane tension decrease was also established in colon cancer cells in the same study". We urge reviewer 1 to show us where this data is documented in Holst et al., since we were unable to find this reference. The only reference to colon cancer cells exists w.r.t 3D invasion assay (there are no experiments in colon cancer cells testing endocytic response to changes in membrane tension) which again is not

in the scope of our manuscript. Furthermore, Holst et al., fail to make any attempt at measuring membrane tension (or surface tension as this reviewer puts it).

We hope that with the above-mentioned arguments it is evident that understanding the role of GRAF1 firstly in CG and CME endocytosis, then going on to show its role in membrane tension regulation is quite a major undertaking by itself. This would require considerable time and more importantly, deflect the main focus of this manuscript.

While it may be indeed difficult to adapt the PDMS-based stretch-relax device to nanotube pulling, several groups have used hypo-iso osmotic cycles to measure the dynamics of membrane tension variations. I disagree therefore that these experiments are beyond the scope of this study as the coupling between endocytosis and membrane tension variations is intrinsically dynamic and is key to the processes investigated here.

Dynamic tether measurement along with modulating tension is an interesting and exciting experiment. This very challenging experiment requires an ability to dynamically modulate tension (either by changing osmolarity or stretching) while being able to maintain tether and measure tether forces. Although this has been shown before¹³, it unfortunately requires extensive modification from a standard optical tweezer set up that we have developed for our purposes.

Tether force measurement is extremely sensitive to various factors such as air draft (air conditioning/person entering the room) and even the fan running in the EM-CCD camera in our hands. Thus holding a thin membrane tether from a cell and measuring forces is an extremely sensitive and difficult process. On top of it, if we need to flow in any medium while holding a membrane tether, we would need to extensively modify our current microscope stage and construct a very precise microfluidic chamber. This is quite a major and extensive technical hurdle that needs to be overcome but at present is beyond the scope of this study. Nevertheless, we are actively trying to build the setup and hopefully the results from it would be part of a future manuscript.

We would like to point out that Reviewer 4 has agreed that dynamical tether force measurement is not feasible at this stage while rest of the three reviewers have not considered dynamic measurement of tether forces necessary for the conclusions made in this paper.

2) Reviewer #2 (Remarks to the Author):

I think that the authors did a great job answering my comments as well as those of my colleagues. The new data on the role of Talin and of the activation of Vinculin are convincing. I also like very much the mechanical/chemical feedback model. It helps understanding how the tension set point could be regulated. There is still work to do to refine the model and understand how to relate time scales to the actual experiments, but I think that at this stage the paper can be published

We thank the reviewer for the support and for recommending the manuscript for publication.

3) Reviewer #3 (Remarks to the Author):

In the revised version of this manuscript, the authors have addressed all of my previous comments satisfactorily. It would have been easier to re-review if the authors would have marked up the changes to the text in a different color.

We apologize for the inconvenience and we thank the reviewer for the support and for recommending the manuscript for publication.

4) Reviewer #4 (Remarks to the Author):

The authors have taken into account some of my comments regarding over-interpreting data, and have argued that the dynamical assessment of tether force that I proposed is not feasible at the moment. I am willing to accept this response.

We thank the reviewer for the critical reading of the rebuttal and agreeing that this finding is worth publication and for concurring that dynamical tether force measurement is not feasible at the moment.

However, upon re-reading the manuscript and the response to reviewer 2 (question 1), it appears to me that there could be a major problem with the interpretation of the data, which I unfortunately overlooked upon my first reading, and which casts serious doubts on the design of the feedback loop shown in Fig.8. The new results linking vinculin activation to Talin, and showing that cells in suspension do not show an increase in endocytosis upon hypo-iso shift are to my mind quite significant. So far, the discussion of this paper relies on a mechano-chemical feedback loop between membrane tension and endocytosis rate. I tend to think that the absence of such feedback in suspended cell suggest that the feedback does directly involve membrane tension, but rather the tension on Focal Adhesion (FA) elements in adherent cells.

We are glad that the reviewer has brought up this issue, since this is a key finding from our work. Since the reviewer was confused in her/his first reading of this manuscript, we have now taken pains to emphasize what we are attempting to say. And have modified the text according to the suggestions made by this reviewer. Please see regions in the text that have been highlighted by yellow highlights.

We completely agree with the interpretation of this reviewer and have incorporated this idea in the discussion (see line 447, page 11). The discovery of the molecular mechanism behind the change in tension triggered endocytic response in attached cells, is indeed a very interesting result. It speaks to the mechano-*chemical* nature of this interplay. Precisely the point we have been trying to convey in our manuscript.

While the two types of tension could be related, and can both be decreased during stretch-release, de-adhesion, or hypo-iso shift experiments, it is not clear at all that there is a direct correlation between FA tension and membrane tension. This suggests that the feedback model (Fig.8b) is missing a key element, which is the tension on the adhesion sites, and which is related in an unclear and potentially complex way to membrane tension in adherent cells. As far as I am concerned, this represents a major difference from the direct feedback loop proposed in this paper. Indeed, one could imagine treatment that would affect the tension on adhesion sites (such as inhibition of myosin contractility) could also lead to up-regulation of the CG endocytotic pathway without relying on obvious variation of the membrane tension itself.

This reviewer is correct his/her/his interpretations : please also see recent manuscripts where researchers have shown that membrane tension could regulate the molecular tension across integrin¹⁴ and regulate adhesion positioning through vinculin¹³ and we have added this in the discussion now (Line 447).

A crucial control in that regard, would be to measure the changes in membrane tension of cells in suspension during a hypo-iso shift. I would expect that membrane tension does indeed decrease in such condition, without triggering the feedback on endocytosis described in this paper, since we already know that such cells do not show the modulation of GC endocytosis. That would show that variation of membrane tension may not be what is measured in the feedback loop, thus invalidating the direct robust feedback proposed in Fig.8.

We agree with the reviewer here. Indeed, when we stretch-relax vinculin null cells, we continue to visualize the passive changes in the membrane morphology as protrusions (reservoirs or VLDs¹⁵) that serve as a proxy for the lowering of tension (see Supplementary Fig. 10c). This result also indicates that change in membrane tension is not sufficient to generate the endocytic response. Further, the plasma membrane of cells in suspension also exhibit similar membrane projections in response to changes in osmotic shock and recovery indicating they do indeed exhibit a lowering of membrane tension in response to the conditions we have utilized (see Movie 3 and montage of images from the videos in Supplementary Fig. 10f). We indicate in our feedback loop that mechanotransduction through vinculin is necessary for the feedback control of CG pathway (See modified Figure 8b).

In fact, as far as I know, it is rather unclear to see at the molecular level how membrane tension, which is a force (per unit length) actin in the plane of the membrane, could modulate the force felt by Talin/Vinculin complexes, which presumably has a sizeable component perpendicular to the membrane.

In the Wang et al paper¹⁴, the authors do make some headway in addressing this vexing point. We now discuss this point in our discussion and refer the readers to this manuscript (line 447; page 11).

These considerations makes me hesitant to recommend publication without discussing the effect of cell adhesion and of FA tension. The fact that the title was amended to specifically restrict the study to adherent cells is good. The paper surely shows a correlation between reduction of membrane tension and increase of CG endocytosis, and this finding is worth publication. However, the causality aspect which is essential to talk about feedback loop has, in my opinion, not been clearly demonstrated. I am conscient of the fact that a discussion has been added on the “real” meaning of tension, which is now called an “apparent tension” that also involves membrane-cytoskeleton adhesion (l.277). I would like to stress that my reservations are of a different nature, and cannot be satisfied by simply stating that Talin/Vinculin tension also contributes to the apparent, or effective tension discussed in this article.

As I realise that changing the focus of the paper to FA tension might be too much to ask, I thus propose that the paper be amended so that all implication of causality between a decrease of membrane tension and an increase of endocytosis are removed, and replaced by a discussion of the existence of a clear correlation between the two quantities. For instance (the list is not exhaustive)

l.74: “we find that a clathrin, caveolin and dynamin-independent endocytic mechanism, the CLIC/GEEC (CG) pathway, rapidly responds to changes in membrane tension, acting to restore it to a specific set point.”

should thus be modified to reflect the actual finding, for instance:

we find that a clathrin, caveolin and dynamin-independent endocytic mechanism, the CLIC/GEEC (CG) pathway, correlates with changes in membrane tension, and might be involved in setting a specific membrane tension set point.”

or

l. 171: “171 Together, these experiments indicated that there is a rapid endocytic response to reduction in membrane tension that is clathrin-, dynamin-, and caveolin-independent.”

should thus be modified, for instance:

there is a rapid endocytic response that correlates with a reduction in membrane tension, but relies on the activation of proteins involved in cell adhesion.

I think this is a crucial point.

We have made the amendments in the paper to reflect the reviewers concerns and indicate correlation between tension modulation of endocytosis. We also include a discussion about membrane tension and tension across focal adhesion. These changes are made at the locations listed here and highlighted in the manuscript as well. (Please see- Line 74, Line 172, Line 183, Line184, Line 237, Line 308, Line 415, Line 430, Line 447.)

minor comment: - Check the sign in Eq.5 and 8 of the Supplementary Material.

We have corrected the sign in the equations in the supplementary material now, and thank this reviewer for her/his vigilant eye.

5) Reviewer #5 (Remarks to the Author):

I have read through the revised manuscript and the replies. The authors have addressed most of my concerns from my previous review.

We thank the reviewer for his or her support and for recommending the manuscript for publication.

References:

1. Pontes, B., Monzo, P. & Gauthier, N. C. Membrane Tension: A Challenging But Universal Physical Parameter in Cell Biology. *Semin. Cell Dev. Biol.* (2017). doi:10.1016/j.semcdb.2017.08.030
2. Lundmark, R. *et al.* The GTPase-activating protein GRAF1 regulates the CLIC/GEEC endocytic pathway. *Curr. Biol.* **18**, 1802–8 (2008).
3. Doherty, G. J. *et al.* The endocytic protein GRAF1 is directed to cell-matrix adhesion sites and regulates cell spreading. *Mol. Biol. Cell* **22**, 4380–9 (2011).
4. Francis, M. K. *et al.* Endocytic membrane turnover at the leading edge is driven by a transient interaction between Cdc42 and GRAF1. *J. Cell Sci.* 4183–4195 (2015). doi:10.1242/jcs.174417
5. Holst, M. R. *et al.* Clathrin-Independent Endocytosis Suppresses Cancer Cell Blebbing and Invasion. *Cell Rep.* **20**, 1893–1905 (2017).
6. Kalia, M. *et al.* Arf6-independent GPI-anchored protein-enriched early endosomal compartments fuse with sorting endosomes via a Rab5/phosphatidylinositol-3'-kinase-dependent machinery. *Mol. Biol. Cell* **17**, 3689–704 (2006).
7. Bitsikas, V., Corrêa, I. R. & Nichols, B. J. Clathrin-independent pathways do not contribute significantly to endocytic flux. *Elife* **3**, e03970 (2014).
8. Kumari, S. & Mayor, S. ARF1 is directly involved in dynamin-independent endocytosis. *Nat. Cell Biol.* **10**, 30–41 (2008).
9. Gupta, G. D. *et al.* Analysis of endocytic pathways in Drosophila cells reveals a conserved role for GBF1 in internalization via GEECs. *PLoS One* **4**, e6768 (2009).
10. Sathe, M. *et al.* Small GTPases and BAR domain proteins regulate branched actin polymerisation for clathrin and dynamin-independent endocytosis. *Nat. Commun.* **9**, 1835 (2018).
11. Sabharanjak, S., Sharma, P., Parton, R. G. & Mayor, S. GPI-anchored proteins are

delivered to recycling endosomes via a distinct cdc42-regulated, clathrin-independent pinocytic pathway. *Dev. Cell* **2**, 411–23 (2002).

12. Gupta, G. D. *et al.* Population distribution analyses reveal a hierarchy of molecular players underlying parallel endocytic pathways. *PLoS One* **9**, (2014).
13. Pontes, B. *et al.* Membrane tension controls adhesion positioning at the leading edge of cells. *J. Cell Biol.* **216**, 2959–2977 (2017).
14. Wang, X. & Ha, T. Defining single molecular forces required to activate integrin and Notch signaling. *Science* (80-.). **340**, 991–994 (2013).
15. Kosmalka, A. J. *et al.* Physical principles of membrane remodelling during cell mechanoadaptation. *Nat. Commun.* **6**, 7292 (2015).

REVIEWERS' COMMENTS:

Reviewer #1 (Remarks to the Author):

I have no further comments about this manuscript.

Reviewer #4 (Remarks to the Author):

In their response to my comments, the authors agreed with my interpretation of their results, in which membrane tension is replaced by mechanical tension on adhesion sites. As I clearly explained in my report, this casts doubts on the existence of a direct mechano-chemical feedback loop between membrane tension and endocytosis, and in particular on the fact that this feedback to fix the homeostatic tension of cells. Indeed, the loop is not closed in their story, since adhesion molecules are also involved. Nevertheless, this idea is still written as such in the paper, and in particular in the abstract of the paper. This needs to be changed, so that the abstract reflects the actual findings and the interpretations of the results that we now agree upon.

To be specific, I have a problem with the sentences:

"We find that vinculin, a well-known mechanotransducer, mediates the tension-dependent regulation of the CG pathway. Vinculin negatively regulates a key CG pathway regulator, GBF1, at the plasma membrane in a tension dependent manner. Thus, the CG pathway operates in a mechanochemical feedback loop with membrane tension, potentially leading to homeostatic regulation of plasma membrane tension."

This sentence is confusing, because "tension-dependent regulation of CG pathway" is used together with "homeostatic regulation of plasma membrane tension." while it is accepted by the author that whatever tension regulates the CG pathway IS NOT the membrane tension. The author must find a way to convey this message in the abstract. They should also remove "feedback loop" from the abstract since the existence of a loop has not been demonstrated. Finally, the abstract must also contain the sentence written by the authors in their response to my comments, but that I couldn't find in the text:

"This result also indicates that change in membrane tension is not sufficient to generate the endocytic response"

Without these changes, I find the abstract misleading.

Other changes that are need:

line 76: change : "is altered upon changes in membrane tension" by "correlates with changes of membrane tension"

line 308 It is not clear what the authors mean by this sentence:

"These results show that, modulating the CG pathway by activating or inhibiting key regulators modifies the membrane tension. Membrane tension, on the other hand, negatively correlates with the operation of the CG endocytic pathway."

Why "on the other hand", while the negative correlation IS A DIRECT CONSEQUENCE of the decrease of membrane tension by CG exocytosis. I would say:

"These results show that, modulating the CG pathway by activating or inhibiting key regulators modifies the membrane tension. This leads to a negative correlation between membrane tension and the operation of the CG endocytic pathway."

Point by Point response to reviewer's comments:

Reviewer #4 (Remarks to the Author):

In their response to my comments, the authors agreed with my interpretation of their results, in which membrane tension is replaced by mechanical tension on adhesion sites. As I clearly explained in my report, this casts doubts on the existence of a direct mechanochemical feedback loop between membrane tension and endocytosis, and in particular on the fact that this feedback to fix the homeostatic tension of cells. Indeed, the loop is not closed in their story, since adhesion molecules are also involved. Nevertheless, this idea is still written as such in the paper, and in particular in the abstract of the paper. This needs to be changed, so that the abstract reflects the actual findings and the interpretations of the results that we now agree upon.

To be specific, I have a problem with the sentences:

“We find that vinculin, a well-known mechanotransducer, mediates the tension-dependent regulation of the CG pathway. Vinculin negatively regulates a key CG pathway regulator, GBF1, at the plasma membrane in a tension dependent manner. Thus, the CG pathway operates in a mechanochemical feedback loop with membrane tension, potentially leading to homeostatic regulation of plasma membrane tension.”

This sentence is confusing, because “tension-dependent regulation of CG pathway” is used together with “homeostatic regulation of plasma membrane tension.” while it is accepted by the author that whatever tension regulates the CG pathway IS NOT the membrane tension. The author must find a way to convey this message in the abstract. They should also remove “feedback loop” from the abstract since the existence of a loop has not been demonstrated. Finally, the abstract must also contain the sentence written by the authors in their response to my comments, but that I couldn't find in the text:

“This result also indicates that change in membrane tension is not sufficient to generate the endocytic response”

Without these changes, I find the abstract misleading.

We thank the reviewer for his/her suggestions and have made the changes. Abstract is now edited as follows.

‘Plasma membrane tension regulates many key cellular processes. It is modulated by, and can modulate membrane trafficking. However, the cellular pathway(s) involved in this interplay is poorly understood. Here we find that, among a number of endocytic processes operating simultaneously at the cell surface, a dynamin independent pathway, the CLIC/GEEC (CG) pathway, is rapidly and specifically upregulated upon a sudden reduction of tension. Moreover, inhibition (activation) of the CG pathway results in lower (higher) membrane tension. However, alteration in membrane tension does not directly modulate CG endocytosis. This requires vinculin, a mechano-transducer recruited to focal adhesion in adherent cells. Vinculin acts by controlling the levels of a key regulator of the CG pathway, GBF1, at the plasma membrane.

Thus, the CG pathway directly regulates membrane tension and is in turn controlled via a mechano-chemical feedback inhibition, potentially leading to homeostatic regulation of membrane tension in adherent cells.’

Other changes that are need:

line 76: change :”is altered upon changes in membrane tension” by ”correlates with changes of membrane tension”

We have edited this paragraph now as follows also to conform to the word limits of the journal (Line 61).

‘Here we have explored the nature of such active responses. We have tested the role of multiple endocytic pathways on modulation of membrane tension by three different approaches. In parallel, we have utilized optical tweezers to measure membrane tension on modulating endocytosis. We find that subsequent to the passive membrane response a clathrin, caveolin, and dynamin-independent endocytic mechanism, the CLIC/GEEC (CG) pathway, is specifically and transiently upregulated. Vinculin, a protein involved in mechano-transduction¹ regulates this tension mediated modulation of endocytosis in adherent cells. In its absence, the CG pathway fails to respond to changes in membrane tension and cell membrane tension is altered. On the other hand, perturbing the CG pathway directly modulates membrane tension, suggesting that this cellular mechanism is likely to be involved in homeostatic control of membrane tension.’

line 308 It is not clear what the authors mean by this sentence:

“These results show that, modulating the CG pathway by activating or inhibiting key regulators modifies the membrane tension. Membrane tension, on the other hand, negatively correlates with the operation of the CG endocytic pathway.”

Why “on the other hand”, while the negative correlation IS A DIRECT CONSEQUENCE of the decrease of membrane tension by CG exocytosis. I would say:

“These results show that, modulating the CG pathway by activating or inhibiting key regulators modifies the membrane tension. This leads to a negative correlation between membrane tension and the operation of the CG endocytic pathway.”

These lines are now edited. Since the first line conveys the conclusion of the experiments from that section, we have deleted the second line to avoid confusion. Now the conclusion for that section is ‘These results show that, modulating the CG pathway by activating or inhibiting key regulators modifies the membrane tension.’ (See Line 296).

Reference:

1. Goldmann, W. H. Role of vinculin in cellular mechanotransduction. *Cell Biol. Int.* **40**, 241–256 (2016).